# Cell-surface receptors enable perception of extracellular cytokinins

Ioanna Antoniadi [1,2], Ondřej Novák [2,3], Zuzana Gelová [4,5], Alexander Johnson [4], Ondřej Plíhal [3,6,7], Radim Simerský [3,8], Václav Mik [3,6], Thomas Vain [2,9,10], Eduardo Mateo-Bonmatí [2], Michal Karady [2,3], Markéta Pernisová [5], Lenka Plačková [3], Korawit Opassathian [1], Jan Hejátko [5], Stéphanie Robert [2], Jiří Friml [4], Karel Doležal [3,6], Karin Ljung [2✉] & Colin Turnbull [1✉]

Cytokinins are mobile multifunctional plant hormones with roles in development and stress resilience. Although their Histidine Kinase receptors are substantially localised to the endoplasmic reticulum, cellular sites of cytokinin perception and importance of spatially heterogeneous cytokinin distribution continue to be debated. Here we show that cytokinin perception by plasma membrane receptors is an effective additional path for cytokinin response. Readout from a Two Component Signalling cytokinin-specific reporter (*TCSn::GFP*) closely matches intracellular cytokinin content in roots, yet we also find cytokinins in extracellular fluid, potentially enabling action at the cell surface. Cytokinins covalently linked to beads that could not pass the plasma membrane increased expression of both *TCSn::GFP* and Cytokinin Response Factors. Super-resolution microscopy of GFP-labelled receptors and diminished *TCSn::GFP* response to immobilised cytokinins in cytokinin receptor mutants, further indicate that receptors can function at the cell surface. We argue that dual intra-cellular and surface locations may augment flexibility of cytokinin responses.

[1] Department of Life Sciences, Imperial College London, London SW7 2AZ, UK. [2] Umeå Plant Science Centre, Department of Forest Genetics and Plant Physiology, Swedish University of Agricultural Sciences, SE-901 83 Umeå, Sweden. [3] Laboratory of Growth Regulators, Institute of Experimental Botany of the Czech Academy of Sciences and Faculty of Science of Palacký University, CZ-78371 Olomouc, Czech Republic. [4] Institute of Science and Technology Austria, 3400 Klosterneuburg, Austria. [5] CEITEC—Central European Institute of Technology and NCBR, Faculty of Science, Masaryk University, CZ-62500 Brno, Czech Republic. [6] Department of Chemical Biology and Genetics, Centre of the Region Haná for Biotechnological and Agricultural Research, Faculty of Science of Palacký University, CZ-78371 Olomouc, Czech Republic. [7] Department of Molecular Biology, Centre of the Region Haná for Biotechnological and Agricultural Research, Faculty of Science of Palacký University, CZ-78371 Olomouc, Czech Republic. [8] Department of Protein Biochemistry and Proteomics, Centre of the Region Haná for Biotechnological and Agricultural Research, Faculty of Science of Palacký University, CZ-78371 Olomouc, Czech Republic. [9] Present address: Chr. Hansen, 2630 Taastrup, Denmark. [10] Present address: Department of Plant and Environmental Sciences, Copenhagen University, 2630 Taastrup, Denmark. ✉email: karin.ljung@slu.se; c.turnbull@imperial.ac.uk

Cytokinins are key hormones regulating cell division and differentiation, root and shoot architecture, senescence and responses to environmental stresses[1,2]. The active forms are the cytokinin free bases, which comprise a range of $N^6$-modified adenine molecules[3,4], especially *trans*-zeatin (*t*Z) and isopentenyl adenine (iP). A cell-level map of cytokinins in roots indicates heterogeneous distribution between different cell types[5]. Homoeostatic regulation of active cytokinin pools occurs at the level of biosynthesis, and also through metabolic deactivation by glucosylation, phosphoribosylation or irreversible degradation by cytokinin dehydrogenase (CKX)[4,6]. Cytokinin signalling commences with perception of bioactive molecules by hybrid histidine kinases (HKs)[1,7,8] that have much lower affinity for cytokinin precursor and glycosyl conjugate forms[9]. Several reports show GFP-fused Arabidopsis HKs (AHKs) mainly localised to the endoplasmic reticulum (ER) membrane[10–12], yet the originally proposed extracellular site of cytokinin perception at plasma membrane receptors[7,13,14] has never been discounted, as highlighted in recent reviews[1,15,16]. Notably, several classes of cytokinin transporters facilitate movement of cytokinins in and out of the cell[16]. Cytokinins binding to receptors trigger a phosphorelay cascade, resulting in activation of B-type Arabidopsis Response Regulators (ARR-B) transcription factors[17], which in turn upregulate type A-ARRs and Cytokinin Response Factors (CRF), a clade of transcription factors within the AP2/ERF superfamily[18]. The cytokinin-responsive synthetic promoter fusion *TCSn::GFP* was developed to reflect global *ARR-B* transcriptional activity[19] and has facilitated in vivo monitoring of cytokinin responses, leading to new discoveries about cytokinin function[20,21]. However, it is unclear whether *TCSn::GFP* signal strength is quantitatively related to cellular cytokinin content and uncertainty remains about which active cytokinin(s) are responsible for different responses in the root tip and other tissues. Here, we used sorted protoplasts to examine the relationship between cell-level *TCSn::GFP* expression and cytokinin content, and also compared intracellular and extracellular cytokinin profiles. The presence of bioactive cytokinins in the latter then led us to test whether cytokinin signalling could operate from receptors at the cell surface. We show that, in addition to the known route of cytokinin signalling via ER receptors, plasma membrane receptors are able to initiate signalling in response to extracellular cytokinins.

## Results

**Cytokinin reporter signal output mirrors cytokinin content.** To test the relationship between endogenous cytokinin content and *TCSn* activity, we analysed cytokinins in root cell protoplasts isolated from *TCSn::GFP* seedlings using fluorescence-activated cell sorting (FACS) (Fig. 1a). Total cytokinin content was almost three times higher in the cytokinin-responsive (GFP$^+$) cells than in the non-responding (GFP$^-$) cells (Fig. 1b). These results are in accordance with evidence for a cytokinin gradient within the root tip[5] that likewise matches the *TCSn::GFP* expression pattern[19]. We further showed a positive correlation between *TCSn::GFP* signal strength and cytokinin content within GFP$^+$ cell subpopulations displaying higher (GFP$^+_{max}$) and lower (GFP$^+_{min}$) mean fluorescence (Fig. 1c and Supplementary Fig. 1). Active cytokinins and their riboside precursors were generally enriched in the more fluorescent cells, whereas inactive cytokinin glucosides were more equally distributed between the two subpopulations (Fig. 1d). We conclude that increases in *TCSn::GFP* readout, designed to approximate global *ARR-B* transcript levels, are indeed associated with elevated active cytokinin content as the input signal. Moreover, *TCSn::GFP* is not simply a binary sensor but instead can report dynamics of changes in cytokinin pool sizes within individual cells.

We further tested the potential of *TCSn::GFP* to respond to perturbations in endogenous cytokinin pool sizes by applying the inhibitor INCYDE (2-chloro-6-(3-methoxyanilino)purine) to block cytokinin degradation by CKX enzymes[22]. Preliminary experiments indicated that the *TCSn* response to INCYDE was stronger than to exogenous cytokinin (Supplementary Fig. 2). In untreated cells, inactive cytokinin glucosyl conjugates were relatively abundant in the GFP$^+$ cells, but *t*Z was the only active cytokinin significantly enriched in this population (Fig. 2a—Mock, light coloured bars). In contrast, iP, which has similarly high affinities to cytokinin receptors[3,9,23], was not enriched. We therefore inferred that *t*Z, rather than iP, has a leading role in the cytokinin response reported by *TCSn*. Indeed, inhibition of turnover by INCYDE resulted in both enhanced *TCSn::GFP* signal (Fig. 2b) and further elevation of *t*Z content in the *GFP$^+$* cells, but had no impact on iP content (Fig. 2a—INCYDE, darker coloured bars).

**Extracellular and intracellular cytokinin profiles differ.** Since cell walls and extracellular space were absent from protoplast samples used in cell sorting (Fig. 2a), we additionally analysed apoplastic and symplastic fractions from roots. LC–MS profiles revealed relative enrichment of cytokinin glucosyl conjugates in the symplast (Fig. 2c), consistent with high levels detected in root protoplasts (Supplementary Fig. 3). However, these conjugate forms are essentially inactive[9] and unlikely to contribute directly to *TCSn::GFP* activation. Glucosyl-conjugate re-conversion to active forms during protoplast isolation was a possibility that was discounted by feeding labelled cytokinins (Supplementary Fig. 4). In contrast, cytokinin free bases and ribosides were either equally distributed between symplast and apoplast or relatively enriched in the latter (Fig. 2c). The presence of bioactive cytokinins in the apoplast led us to hypothesise that extracellular cytokinins could potentially initiate signalling.

**Extracellular cytokinins can activate cytokinin signalling.** Based on finding cytokinins in the apoplast, we next tested whether the bioactive compounds could be perceived by plasma membrane receptors[1,7,13–16,24], by treating *TCSn::GFP* protoplasts with iP or *t*Z in free solution or covalently attached to Sepharose beads via flexible linkers designed to minimise steric hindrance to cytokinin binding (Supplementary Fig. 5). Since the beads are much larger than the protoplasts (Supplementary Fig. 6a, red arrows), the attached cytokinin ligands were unable to enter the protoplast, and could thus be considered as membrane-impermeant signals. *TCSn* fluorescence signal strength after treatment with bead-bound cytokinins provided in vivo evidence for activation of cytokinin response through perception of extracellular cytokinins (Fig. 3a, b, also Supplementary Fig. 7b for *TCS::GFP* response). Further analysis showed that ~0.2–0.6% of cytokinins with their linkers and up to 0.2% of the free cytokinin ligands had potentially been detached from the beads (Fig. 3c, Supplementary Table 1). As predicted from their N9 substitutions, the cytokinins with linkers have substantially lower bioactivity both in TCSn activation (Supplementary Fig. 8a) and in receptor binding (Supplementary Fig. 8d) experiments. Because of their lower activity and because attachment internally to the bead matrix will hinder ligand access to the protoplast surface, we compensated by using a moderate excess of immobilised ligands (calculated as net mean density of $10 \, \mu mol \, l^{-1}$) compared with free cytokinin concentrations of 2 μM. The quantities of detached cytokinin with linker would not lead to significant TCSn expression, and minimal conversion of cytokinin with linkers to free cytokinins was detected during incubations with root tissues (Supplementary Fig. 8b, c). For the free cytokinins, the detached amounts

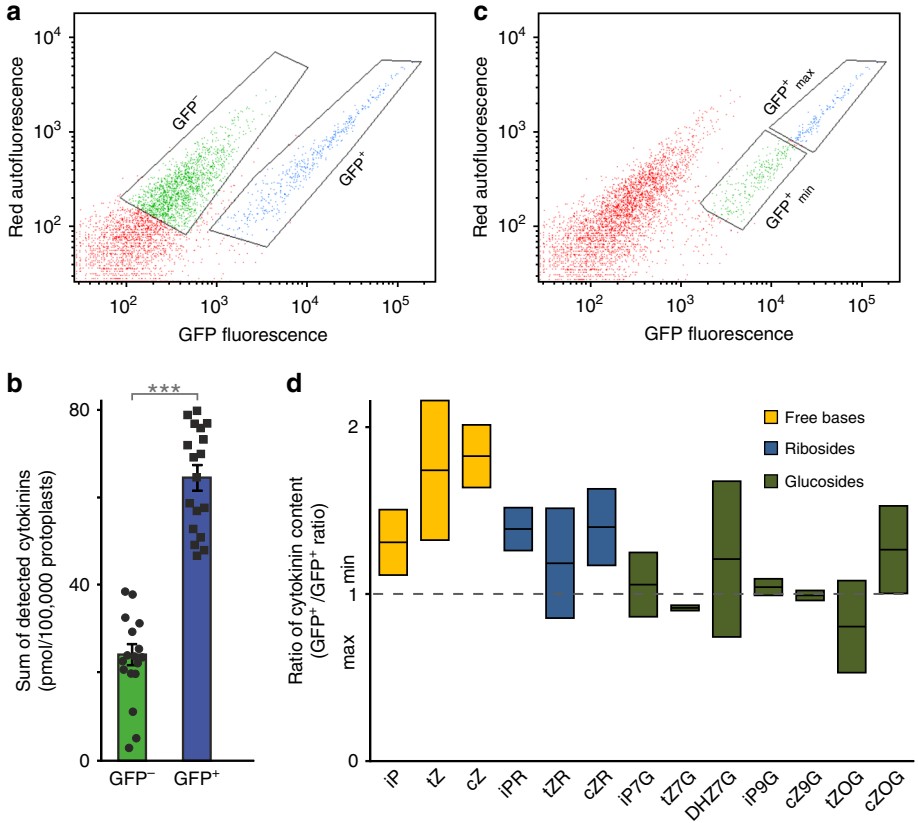

**Fig. 1 Cell-level cytokinin content correlates with expression of the *TCSn::GFP* cytokinin reporter gene.** Analysed by fluorescence-activated cell sorting (FACS) and liquid chromatography–mass spectrometry (LC–MS). **a** Autofluorescence scattering intensity plotted against GFP fluorescence intensity for 50,000 events analysed, separating the *TCSn::GFP* root protoplasts into two populations, GFP− cells (green) and GFP+ cells (blue) which were then selected (gated) for cell sorting. **b** Sum of detected cytokinin metabolites in sorted GFP+ and GFP− cells, individual values as dot plot with bar (mean with s.e.m). Paired sample *t*-test applied (***$p < 0.001$, $n = 9$ biological replicates, two technical replicates for each biological replicate). **c** As for **a**, but showing selection of *TCSn::GFP*+ cell sub-populations with maximum fluorescence (GFP+max; blue) and minimum fluorescence (GFP+min; green). **d** Ratios of the concentration of cytokinin metabolites between *GFP*+max and GFP+min cells. Whiskers indicate entire range of values, central line is mean; $n = 2$. Bar colours represent different cytokinin metabolite groups: free bases (yellow), ribosides (blue), glucosides (green). All protoplast samples derived from 9-day-old *Arabidopsis* seedling roots. See also Supplementary Fig. 1.

correspond to 4 nM free iP or *t*Z (Fig. 3c and Supplementary Fig. 9), concentrations that likewise would not lead to a significant *TCSn::GFP* response (Fig. 3d), let alone the very large responses seen with both free and immobilised cytokinins. Notably, extracellularly restricted *t*Z resulted in a *TCSn::GFP* response approaching that elicited with 2 μM free *t*Z, whereas the response to immobilised iP was substantially lower (Fig. 3a, b). These results suggest that apoplastic *t*Z can trigger cytokinin response, consistent with the increase in *TCSn* signal and *t*Z level when cytokinin degradation is impaired (Fig. 2a, b).

We further explored functional importance of signalling from extracellular cytokinins by analysing expression of members of the CRF clade within the AP2/ERF transcription factor superfamily, several of which are regulated via the canonical TCS signalling pathway[18]. Transcript abundance of *CRF6* measured by qPCR was significantly enhanced by both extracellular and free iP and *t*Z (Fig. 3g), whereas *CRF3* was upregulated by both free cytokinins and by extracellular iP, but not by extracellular *t*Z (Supplementary Fig. 7e).

Although iP, unlike *t*Z, was not enriched in the cytokinin-responsive cells (Fig. 2a) nor in the apoplast (Fig. 2c), exogenous supplies of both compounds triggered cytokinin responses in protoplasts (Fig. 3a, b), consistent with previous studies[25,26]. Cytokinin treatment of whole seedlings indicated similar significant enhancement of *TCSn* response by iP and *t*Z in roots, but

a differential spatial regulation was also observed (Fig. 3e, f; also Supplementary Fig. 7a for corresponding *TCS::GFP* responses). iP had the strongest effect on meristematic stele cell initials (3–28 μm in stele initials; Fig. 3e, f, highlighted in magenta), whereas *t*Z response was maximal in the transition zone in the stele (221–230 μm from stele initials; Fig. 3e, f, highlighted in green). We confirmed that minimal conversion of exogenous iP to *t*Z occurred (Supplementary Table 2) and therefore the response to iP was not due to increased *t*Z levels. Although the exact biological role for cytokinins in the stele initials remains to be determined, *TCS* signal in those cells was absent in *ahk4* mutant roots[27] (Supplementary Fig. 7c, d) providing genetic evidence for AHK4 being essential in cytokinin perception in the stele.

**Receptor dependence of extracellular cytokinin signalling.** To evaluate whether the *TCSn* response to extracellular cytokinins acted through one or more cytokinin receptors, we tested each of the three AHK receptors individually in the presence or absence of extracellular and free cytokinins, using *TCSn::GFP* lines mutated in the other two *AHKs*[27]. Although absolute signal strength was diminished in these mutant lines compared with the wild type, as shown elsewhere[25,27], they all retained responsiveness to free IP and *t*Z[26] (Fig. 4a, b). In root protoplasts, equivalent levels of response to both extracellular cytokinins were found for

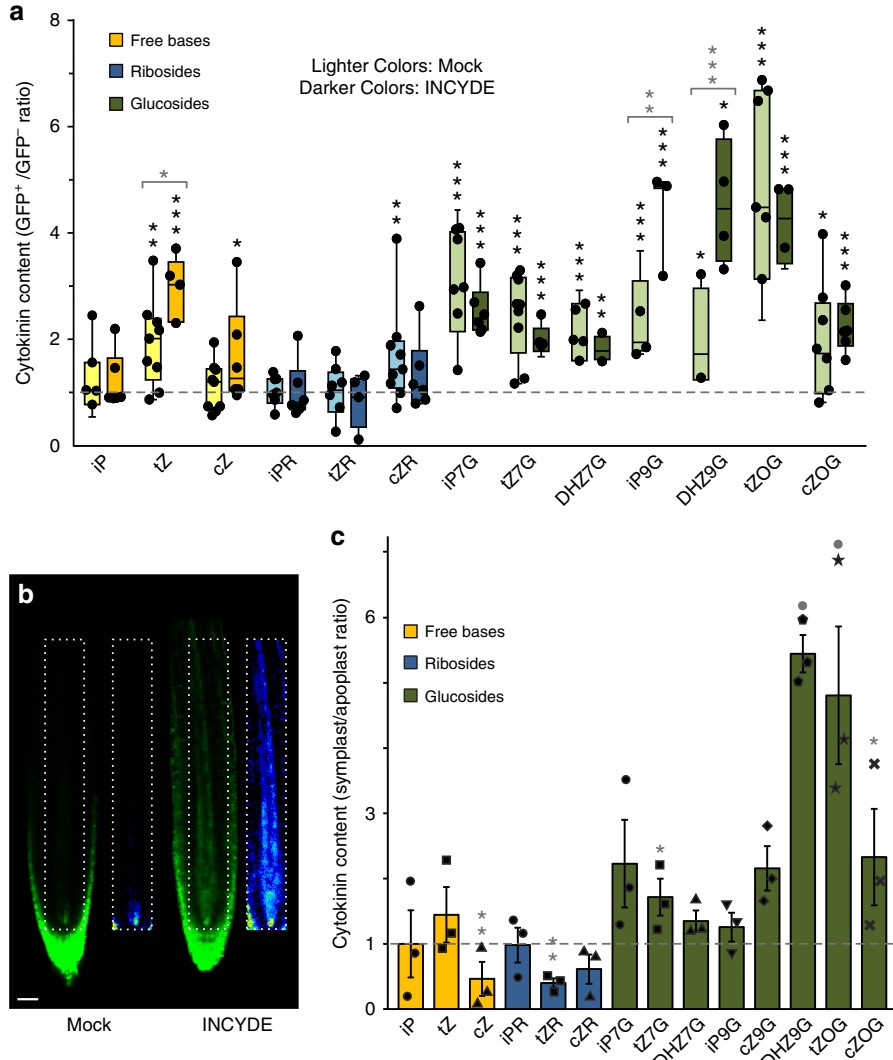

**Fig. 2 Cytokinin response correlates with enrichment of intracellular *trans*-zeatin content and is enhanced by inhibition of cytokinin turnover, but bioactive extracellular cytokinins are also present. a** Ratio of cytokinin metabolite concentration between GFP$^+$ and GFP$^-$ protoplasts from 9-day-old *TCSn::GFP* seedlings roots, treated with or without 20 μM INCYDE during protoplasting. Cytokinins were quantified as fmol per 100,000 protoplasts. Colours represent different cytokinin metabolite groups: free bases (yellow), ribosides (blue) and glucosides (green). Darker bars are INCYDE-treated samples and lighter colours are corresponding mock treatment. Whiskers represent the entire range of values, boxes indicate first and third quartiles, centre line is median, dots are individual values; $n = 9$ for mock and $n = 6$ for treated samples. Black asterisks indicate statistically significant differences in cytokinin concentration between GFP$^+$ and GFP$^-$ cells of *TCSn::GFP* treated or mock samples (paired sample *t*-test). *$p < 0.05$; **$p < 0.01$; ***$p < 0.001$. Grey asterisks above brackets denote statistically significant difference in cytokinin ratios between mock and treated experiment (one-way ANOVA and Tukey's test, significance levels as above). **b** Confocal imaging of *TCSn::GFP* 6-day-old roots with or without 10 μM INCYDE treatment for 6 h. Insets show the respective root vasculature in 16-colour LUT (Look-Up Tables; https://imagej.net/) highlighting the gradations of fluorescence intensity. Scale bar is 35 μm. **c** Detection of extracellular cytokinins: ratios of cytokinin concentrations in symplastic/apoplastic fluid extracted from 9-day-old *Arabidopsis* wild-type roots. Ratios were derived from cytokinin levels calculated as fmol (g fresh weight)$^{-1}$ of original tissue and are shown as scatter plot with bar (mean with s.e.m). $n = 3$ pools of at least 1500 roots; colour coding as in **a**. Grey asterisks or dots indicate statistically significant differences in cytokinin concentration between symplast and apoplast by paired sample *t*-test. Significance levels are: dot, $p < 0.1$; *$p < 0.05$; **$p < 0.01$.

AHK2 (*ahk3 ahk4*), but interestingly AHK3 (*ahk2 ahk4*) and AHK4 (*ahk2 ahk3*) responded only to apoplastic IP or *t*Z, respectively. The application of IP and *t*Z to whole seedlings of the respective genotypes showed that AHK4 is not only essential, but also sufficient for cytokinin response in the stele, consistent with previous findings[26] (Fig. 4b). In contrast, the same treatments of seedlings carrying only AHK3 or AHK2 receptors resulted in slightly enhanced *TCSn::GFP* response in some cell files within the stele (Fig. 4b bottom panels—vasculature). The baseline expression of *TCSn::GFP* in columella cells is constitutively high and did not show obvious further increases in response to exogenous cytokinins. As columella cells contain higher levels of endogenous cytokinins than found in other root tip cell types[5], it may be that the columella cytokinin content is non-limiting or saturated in terms of *TCSn* signalling. Visualisation of AHK4–GFP and AHK3–GFP fusion proteins by 3D AiryScan microscopy indicated that a proportion of both cytokinin receptors was not co-localised with ER (Fig. 4c, Supplementary Fig. 10). Similar to previous[11,12] and recent[28] reports, some of the non-ER AHK signal was clearly at the cell surface,

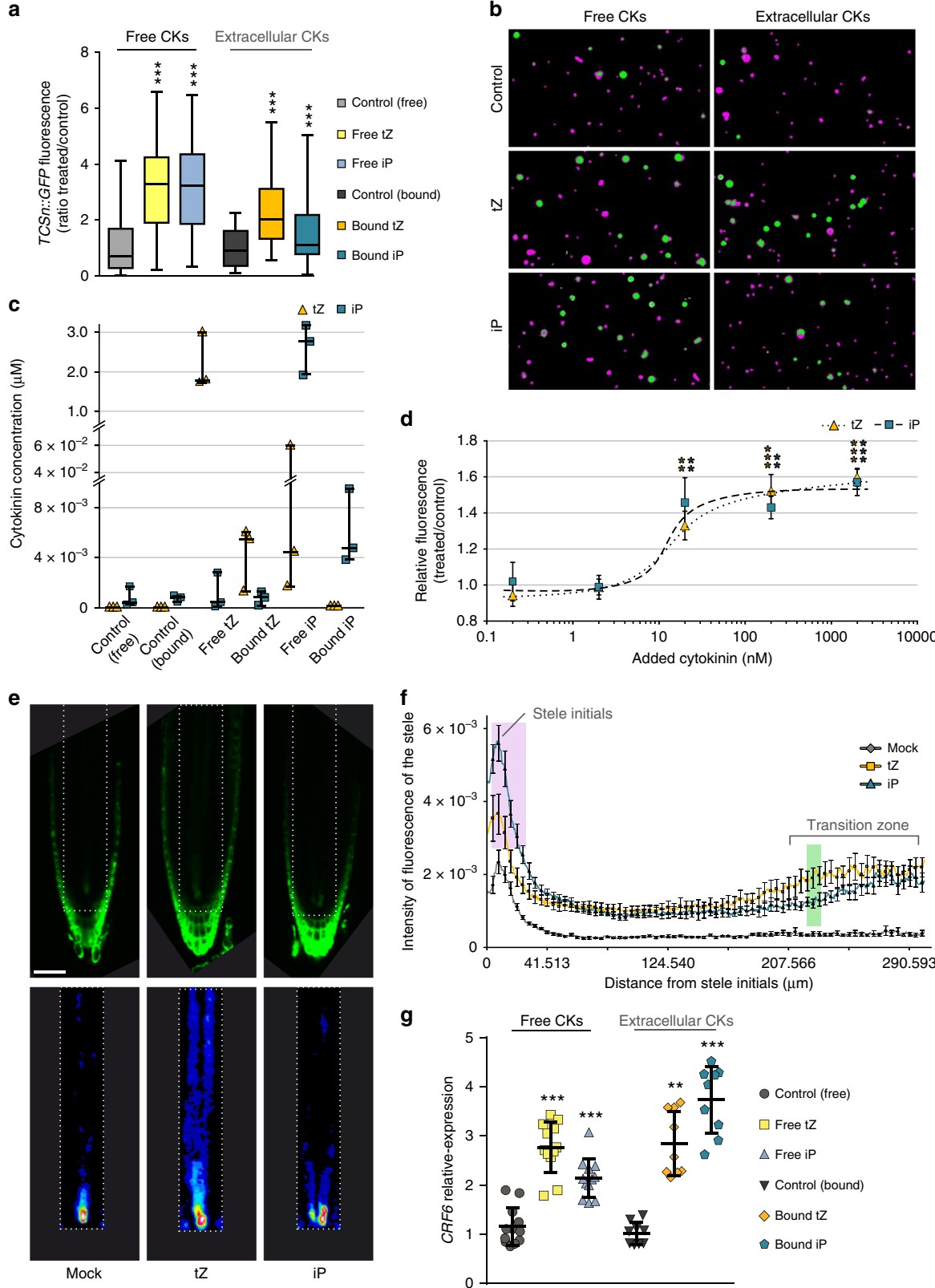

indicating plasma membrane localisation. Across multiple imaged cells, 25% of AHK3 and 36% of AHK4 signals were localised in non-ER regions (Fig. 4d). These imaging experiments further support our functional evidence showing that extracellular cytokinins can be perceived by the sub-populations of AHK receptor proteins that reside on the cell surface.

## Discussion

Our initial aim in this work was to explore the relationship between output strength of the *TCSn* cytokinin reporter and input signal level in terms of cytokinin content of individual cells. It can be argued that changes in *TCSn* signal could precisely reflect fluctuations in bioactive cytokinin levels, but equally it

**Fig. 3 Extracellular cytokinins activate cytokinin responses. a** Quantification of GFP fluorescence in protoplasts, derived from roots of 6-day-old *TCSn::GFP* seedlings, after treatment with or without free cytokinins (*tZ* or iP, 2 μM, denoted "free") or immobilised cytokinins (*tZ* or iP, ligand mean density 10 μmol l$^{-1}$ equivalent, attached to Sepharose beads, denoted "bound", also referred to as extracellular compounds). Negative controls without added cytokinin were incubations with and without beads (control free and bound, respectively). Whiskers represent the entire range of values, boxes indicate first and third quartiles, centre line is median); ***$p < 0.001$ by one-way ANOVA and Tukey's test, indicating significant differences in fluorescence intensity between control and corresponding free or extracellular cytokinin treatments. Three independent experiments were performed, with each comprising data from $n > 20$ images, corresponding to >1000 protoplasts. Cytokinin treatment was for 16 h then 1 μM FM4-64 applied 5 min prior to confocal imaging. See also Supplementary Figs. 5–7b. **b** Images of the treated protoplast samples described in **a**. GFP (green) and FM4-64 (magenta) channels are overlaid. **c** Quantification of iP and *tZ* in the remaining protoplast samples after the 16 h cytokinin treatment from the experiments described in **a**. Dots are individual values; whiskers represent the entire range of values, boxes indicate first and third quartiles, centre line is mean. **d** Dose-response of GFP intensity in protoplasts of *TCSn::GFP* root protoplasts treated with 0.2, 2, 20, 200 or 2000 nM free *tZ* or iP for 16 h. Data are shown as ratio of treated/control. Error bars are s.e.m. and asterisks indicate significant differences in *TCSn::GFP* response (**$p < 0.01$; ***$p < 0.001$ by homoscedastic *t*-test). Other details as for **a**. **e, f** Responses of 6-day-old *TCSn::GFP*-expressing roots to treatment with 2 μM *tZ* or iP for 16 h. **e** Confocal images of roots, scale bar 54 μm. Top panels are GFP intensity signal; bottom panels show the respective root vasculature in 16-colour LUT (Look-Up Tables; https://imagej.net/) to better highlight regions with enhanced cytokinin response. **f** Quantification of GFP fluorescence intensity in the stele ($n \geq 11$). Results are combined from two independent experiments, with the fluorescent signal of ten roots quantified in each experiment using ImageJ. Error bars are s.e.m. Both cytokinins significantly enhanced *TCSn::GFP* response compared with controls, while the highlighted areas in magenta (iP > *tZ*) and green (*tZ* > iP) show significant differences between *tZ* and iP responses ($p < 0.05$, homoscedastic *t*-test). **g** qRT-PCR analysis of expression of *Cytokinin Response Factor 6* (*CRF6*) induced by free and extracellular bound cytokinins (experimental setup as in **a**). Dots are individual values, whisker represents s.d., centre line is mean. Data are relative expression by the $2^{-\Delta\Delta Ct}$ method. Asterisks indicate values significantly different from the corresponding control treatment (**$p < 0.01$; ***$p < 0.001$ by Mann–Whitney *U* test). Four biological replicates were analysed in triplicate.

could be that *TCSn* output is modified through variation in abundance of receptors or downstream signalling components. The evidence we provide here is that root cells with significant *TCSn::GFP* expression contain nearly threefold more total cytokinin than cells sorted into the GFP$^-$ category. Moreover, when we sorted the GFP$^+$ cells into two pools representing higher and lower GFP signal, there was again a trend towards higher cytokinin in the GFP$^+_{max}$ than in the GFP$^+_{min}$ cells. When cytokinin catabolism was blocked with the inhibitor INCYDE, *TCSn::GFP* signal in whole roots was greatly enhanced especially in stele cells, and protoplasts showed higher levels of *tZ*, but not of the other bioactive cytokinins *cZ* and iP. Although we did not attempt to provide a full calibration curve, it is clear that *TCSn* behaves as a cytokinin sensor with quantitative characteristics, and *tZ* is the bioactive cytokinin whose level most closely tracks the *TCSn* output.

In addition to assessing signalling strength from intracellular cytokinins, we also found substantial pools of extracellular cytokinins, leading us to test whether cell surface perception could occur. Several reports using AHK translational fusions to fluorescent proteins point to AHK receptors predominantly located on ER membranes[10,11], and show that AHK interaction with their downstream AHP partners can also occur at the ER[12]. However, plasma membrane-localised receptors have never been excluded[11,29], and AHK3 and AHK4 have been shown to at least partially reside in the plasma membrane[10,11]. Although we did not attempt to make detailed quantitation of relative proportions of AHKs on plasma membrane vs ER, our image analysis likewise clearly indicates that significant amounts of AHK are found in non-ER regions including the cell surface. Our data are strongly corroborated by independent experiments showing co-localisation of plasma membrane markers both with AHK4–GFP and with a fluorescently tagged iP derivative that preferentially binds AHK4[28]. Moreover, Kubiasová et al.[28] show that AHK4 and tagged iP are found in brefeldin A-dependent endocytic vesicles characteristic of trafficking to and from the plasma membrane. Despite the strong evidence for ER location of AHKs, recent reviews have highlighted the lack of direct substantiation of extracellular cytokinin perception[1,15,16]. In this context, our multiple strands of evidence for responses to extracellular cytokinins initiated via plasma membrane-bound receptors indicates that both sites of perception appear to exist. Previous studies

have also detected a wide range of extracellular cytokinins[30,31]. However, the relative abundance of different forms varies substantially, possibly because of taxonomic and experimental differences.

In particular, our in vivo data point to perception of apoplastic *tZ* being an important route for cytokinin response activation in roots (Figs. 2c, 3a and 4a), while its symplastic degradation might act as a negative feedback loop in cytokinin signalling (Fig. 2a). These results are consistent with evidence showing that impaired cytokinin import/uptake results in induction of cytokinin response[24] and with *tZ*-specific binding by AHK4 in outer membranes[11]. Moreover, selective extracellular degradation by CKX expression leads to diminished cytokinin responses, whereas intracellular targeting of CKX did not have such an effect[24]. Nonetheless, intracellular inactivation of *tZ* by endogenous CKX does appear to occur (Fig. 2a—INCYDE)[32].

The spatially distinct tissue-level responses to iP and *tZ* (Fig. 4a, b) may relate to ligand preferences[3,9,23], sites of maximal expression[8] of each AHK type or/and differentially localised and expressed CKX enzymes[33]. Indeed, given that there is so much heterogeneity between root cell types, the distribution of ER vs non-ER AHK proteins in the example cells shown in Fig. 4c will likely differ if other cell types are examined. Given that *tZ* cytokinins have previously been suggested to be dispensable for root function[34], our evidence for a dominant role of *tZ* rather than iP is surprising. Moreover, although Kubiasová et al.[28] did not test *tZ*, they show that a tagged version of iP can enter the secretory pathway, resulting in delivery to the apoplast. One explanation for the discrepancy could be that endogenous and exogenous iP compounds behave differently, for example, if the latter is present at non-physiological concentrations or is distributed into abnormal cellular compartments.

We found contrasting-specific extracellular cytokinin responses in different *ahk* double mutant protoplasts, where only a single AHK type remained functional. The response strengths to *tZ* and iP did not necessarily correspond to the known affinities of each AHK for these ligands[3,9,23]. One reason for the lack of correspondence may be that each AHK here is acting only as a homodimer, whereas in wild type plants heterodimerisation may occur[12]. Although localisation-related selectivity of receptor–ligand interactions merits further exploration, here we have demonstrated the presence of functional receptors at the plasma membrane of the

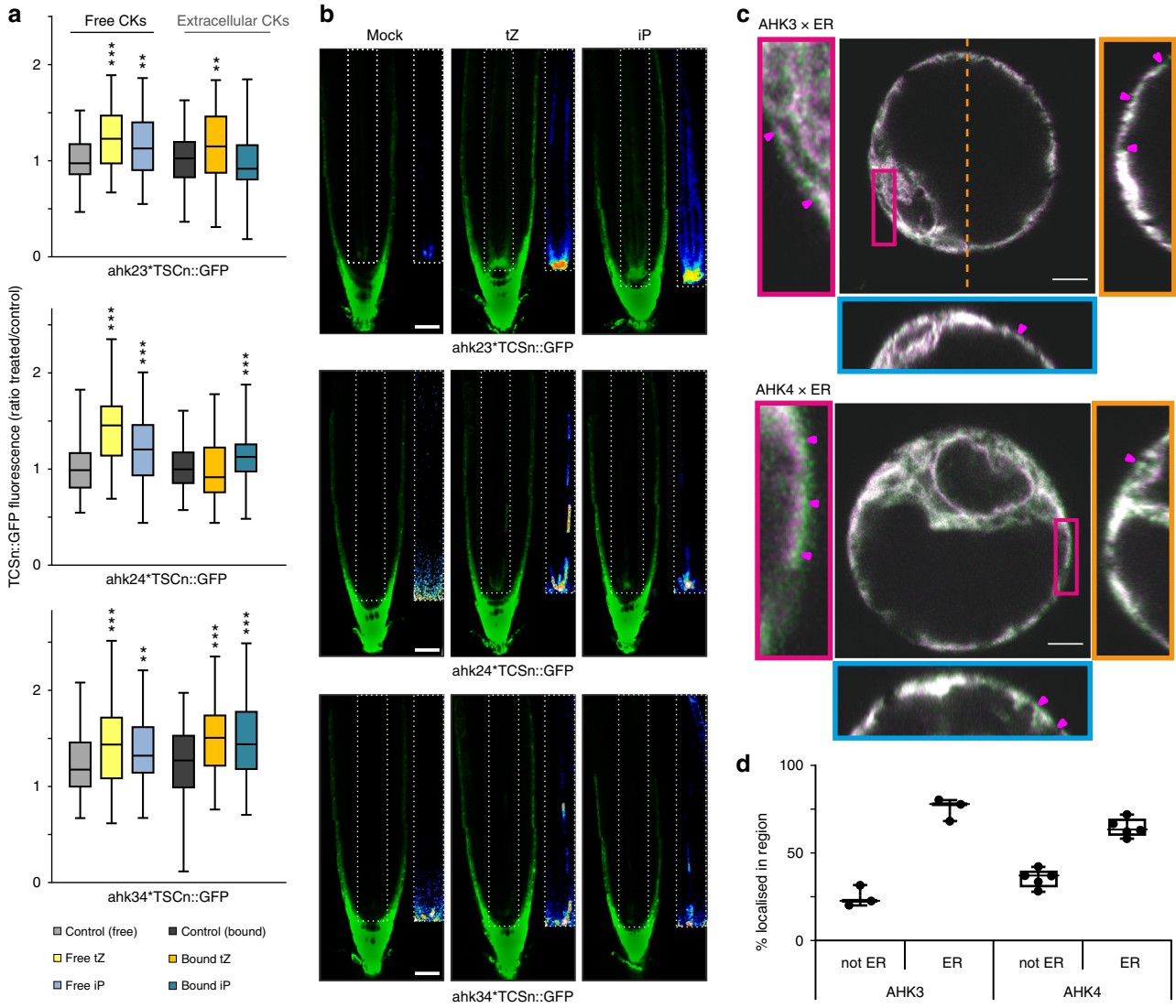

**Fig. 4 Cytokinin responses when only one AHK receptor is active, and receptor localisation. a** Quantification of GFP fluorescence in protoplasts derived from roots of 6-day-old *TCSn::GFP* seedlings in wild type (Col-0), *ahk3,4*, *ahk2,3* and *ahk2,4* backgrounds, after treatment for 16 h with or without free cytokinins (*tZ* or iP, 2 μM, denoted "free") or immobilised cytokinins (*tZ* or iP, ligand mean density 10 μmol l⁻¹ equivalent, attached to Sepharose beads, denoted "bound"). Negative controls without added cytokinin were incubations with and without beads (control free and bound, respectively). Whiskers indicate entire range of values, box indicates first and third quartiles, and central line is mean; **$p < 0.01$; ***$p < 0.001$ by one-way ANOVA and Tukey's test, indicating significant differences in fluorescence intensity between control and corresponding free or extracellular cytokinin treatments. Three independent experiments were performed with each comprising data from $n > 20$ images, corresponding to >1000 protoplasts. See also Supplementary Figs. 7 and 10. **b** Confocal images of roots from *ahk3,4*, *ahk2,3* and *ahk2,4* double mutants expressing *TCSn::GFP*, after treatment for 24 h with 100 nM *tZ* or iP. Scale bar is 39 μm. The inset panels show the respective root vasculature in 16-colour LUT (Look-Up Tables; https://imagej.net/) highlighting the gradations of fluorescence intensity. **c** 3D Airyscan images of Arabidopsis protoplasts expressing AHK3–GFP (green; above) and AHK4–GFP (below) and the ER maker RFP-p24δ5 (magenta). The left panel shows a zoomed in region of the cell (red rectangle). The right panel depicts the YZ orthogonal view of the Z-stack (orange dashed line). The bottom panel shows the XZ orthogonal view of the Z-stack (blue dashed line). Magenta arrows indicate regions of AHK-only signal on the cell surface. Scale bars, 5 μm. **d** Quantitative analysis of AHK-GFP signal co-localising or not co-localising with ER marker RFP-p24δ5. Whiskers represent entire range of values, boxes indicate first and third quartiles, central line is median and dots are values for individual cells. $n = 3$ for AHK3, $n = 5$ for AHK4.

cell, thus substantially resolving the lengthy debate on this issue[1,11,12,15,16,24,28,29]. Dual location of receptors may potentially provide plants with additional flexibility in cytokinin responses. It remains to be ascertained whether different biological functions are associated with each location. Overall, many regulatory elements can influence cytokinin response including selective molecule-receptor affinities and influences of tissue-specific apoplastic pH[35], together with heterogeneous tissue- and cell-specific distribution of

cytokinins, their cognate receptors and cytokinin inactivation enzymes.

## Methods

**Plant material and growth conditions**. All experiments used Arabidopsis (*Arabidopsis thaliana*). The transgenic line *TCSn::GFP*[19] was used for all cell-sorting experiments, *TCSn::GFP* and *TCS::GFP*[25] were employed in protoplast and seedling treatment experiments, and the ecotype Col-0 was used for extraction of apoplast/

symplast and for feeding experiments with labelled cytokinins. Crossed lines *ahk x TCSn::GFP* were published previously[27] and the mutant lines *cre1-2 (ahk4)*[8], *ahk2-5*, *ahk3-7*, *ahk2-5 ahk3-7 (ahk2,3)*, *ahk2-5 cre1-2 (ahk2,4)* and *ahk3-7 cre1-2 (ahk3,4)*[36] were crossed also with *TCS::GFP*. Homozygous lines were then identified and used in confocal experiments. In all experiments, the seeds were surface sterilised with 20% (v/v) dilution of bleach for 5 min (2 × 2.5 min) and then rinsed five times with sterile water. For cell-sorting experiments, the seeds were sown in three rows (100 seeds/row) on square Petri dishes containing standard Murashige and Skoog medium (4.4 g/l Murashige and Skoog salt mixture, 1% sucrose, 0.5 g/l MES, 1% agar and adjusted to pH 5.7 with KOH), covered with sterile mesh squares to facilitate the harvesting of the apical part of the primary root, and were stratified in darkness at 4 °C for 3 days. Seedlings were then grown on plates placed vertically, for 9 days at 23 °C under 150 μmol m$^{-2}$ s$^{-1}$ light with photoperiod of 16-h light and 8-h darkness. One standard cell-sorting experiment required 20 Petri dishes. For confocal microscopy experiments, sterilised seeds were sown (10 seeds/row), stratified and grown for 6 days as described above.

**Fluorescence-activated cell sorting (FACS).** The distal half of 9-day-old roots of *TCSn::GFP* seedlings were harvested, protoplasts were isolated in the presence or absence of 20 μM INCYDE and sorted using a BD FACS Aria I flow cytometer (BD Biosciences)[5,37,38]. The software used for data processing was BD FACSDiva version 6.1.2. Isolated protoplasts were loaded in the cell sorter (4 °C) and passed individually through a 100 μm nozzle (Becton Dickinson and Company) using 0.7% NaCl as sorting buffer with sheath pressure 20 psi. Healthy protoplasts were initially selected using a 488 nm (blue) laser for excitation of their autofluorescence. For analysis of protoplasts' relative size, the forward scatter detector was used while and for their respective granularity and complexity analysis, the side scatter detector was employed (bandpass filter 488/10). The healthy protoplasts were additionally interrogated for emission of GFP fluorescence after excitation with the blue laser. The gating of the GFP$^+$ and GFP$^-$ populations took place in a bi-plot of GFP fluorescence (bandpass filter 530/30, dichroic mirror 520) and autofluorescence emission (bandpass filter 610/20, dichroic mirror 595) of the healthy protoplasts. The gated populations were sorted, then frozen immediately in liquid nitrogen and stored at −80 °C until purification. From loading into the FACS until reaching the fully isolated GFP$^+$ and GFP$^-$, or GFP$^+_{max}$ and GFP$^+_{min}$ populations (~2 h), protoplasts remained at 4 °C. Each biological replicate represents an independent experiment.

**Protoplast isolation.** Protoplasts were isolated from roots of 9-day-old *Arabidopsis* seedlings[5,37,38]. The tissue was excised and rinsed in a 40-μm cell strainer (BD Falcon) with distilled water. The harvested tissue was then submerged into the protoplast isolation buffer (600 mM mannitol, 2 mM MgCl$_2$, 10 mM KCl, 2 mM CaCl$_2$, 2 mM MES and 0.1% BSA, pH 5.7) supplemented with pectolyase (0.3 units mL$^{-1}$) and cellulysin (45 units mL$^{-1}$). After 2 h incubation at 22 °C in darkness and with gentle stirring at 46 rpm, the protoplasts were isolated using a 40-μm cell strainer. Then they were centrifuged for 3 min at 1000 *g* at 4 °C and the resulting protoplast pellet was resuspended in 1 mL of cold sorting buffer (0.7% NaCl) and kept at 4 °C until further processing.

**Protoplast feeding experiments.** One μM of [$^{13}C_5$] *tZ* or [$^{13}C_5$]*cis*-zeatin (*cZ*) was added to all protoplast samples except samples comprising the 0-min time-point, which were harvested immediately; centrifuged at 1000 *g* for 3 min at 4 °C, then frozen in liquid nitrogen and stored at −80 °C until cytokinin analysis. After 30, 60 and 90 min of incubation in the dark at room temperature and under continuous shaking (46 rpm), treated samples were similarly collected. Cell numbers per sample were estimated by hemocytometer.

**Isolation of apoplastic and symplastic fractions.** Apoplastic and symplastic fluids were isolated from roots of 9-day-old Col-0 seedlings[39]. The tissue was harvested, rapidly weighed and positioned in a 1 ml syringe without plunger. The apoplastic fraction was recovered by placing the syringe containing the sample roots in a 15 ml centrifuge tube which was then centrifuged at 900 *g* for 20 min at 4 °C. The syringe containing the remaining root tissue was wrapped in aluminium foil, rapidly frozen in liquid nitrogen and then allowed to thaw at room temperature. Finally, the syringe was placed in a clean 15 ml centrifuge tube into which the symplastic fluid was collected by 15 min centrifugation at 2500 *g* at 4 °C. Collection of apoplastic fluid by this protocol has been shown to contain little cytoplasmic contamination[39,40]. Here, the presence of symplastic fluid in the apoplastic fractions was assessed to be 10–15%, based on assays using the cytosolic malate dehydrogenase enzyme marker. The different cellular origins of the two fractions were further confirmed by the resultant contrasting cytokinin profiles of the "apoplast" and "symplast" shown in Fig. 2c. The cytokinin profile of the root symplast (Fig. 2c) was noted to be highly similar to the profile of the root protoplasts (Fig. 2a, Supplementary Fig. 3). Three biological replicates were analysed and each was a pool of at least 1500 seedlings.

**Cytokinin quantification.** After frozen samples were thawed on ice and mixed in 1 ml of 1 M formic acid together with a cocktail of stable isotope-labelled internal standards used as a reference (0.25 pmol of CK bases, ribosides, N-glucosides, and 0.5 pmol of CK O-glucosides, nucleotides per sample added). Cytokinins were purified using in-tip solid-phase microextraction based on the StageTips technologyas as described previously[41]. Briefly, combined multi-StageTips (containing C18/SDB-RPSS/Cation-SR layers) were activated sequentially with 50 μl each of acetone, methanol, water, 50% (v/v) nitric acid and water (by centrifugation at 434 *g*, 15 min, 4 °C). After application of the sample (500 μl, 678 *g*, 30 min, 4 °C), the microcolumns were washed sequentially with 50 μl of water and methanol (525 *g*, 20 min, 4 °C), and elution of samples was performed with 50 μl of 0.5 M NH$_4$OH in 60% (v/v) methanol (525 *g*, 20 min, 4 °C). The eluates were then evaporated to dryness in vacuo and stored at −20 °C. The cytokinin profile was then quantitatively analysed by multiple reaction monitoring UHPLC-MS/MS (1290 Infinity Binary LC System coupled to a 6490 Triple Quad LC/MS System with Jet Stream and Dual Ion Funnel technologies in positive ion mode; Agilent Technologies), as described previously[5]. Cytokinin concentrations were determined using Mass Hunter software (version B.05.02; Agilent Technologies) using stable isotope dilution. Labelled and endogenous cytokinin metabolites after protoplast treatment with $^{13}C_5$-*tZ* and $^{13}C_5$-*cZ* were also isolated by multi-StageTips and measured using LC–MS/MS, with the multiple reaction monitoring transitions as described above.

**Seedling treatments.** Six-day-old *TCSn::GFP* or *TCS::GFP* seedlings were transferred into 6-well plates containing 2 ml of standard Murashige and Skoog liquid medium with additions of 10 μM INCYDE, 10 μM 6-benzylaminopurine (BAP), 2 μM *tZ* or 2 μM iP. After 6 h, samples treated with INCYDE or BAP were collected for confocal imaging, whereas samples treated with *tZ* and iP were similarly examined after 16 h. During incubation, samples were placed on an orbital shaker at 126 rpm under normal growth conditions. Seedlings of cytokinin receptor mutants (*ahk2,3*, *ahk3,4* and *ahk2,4* in *TCSn::GFP* background) were transferred for 24 h to solid media containing 100 μM IP or *tZ*.

**Protoplast treatments.** Protoplasts were isolated from 6-day-old roots of *TCSn::GFP, TCS::GFP, TCSn::GFP ahk2,3, TCSn::GFP ahk2,4* and *TCSn::GFP ahk3,4* seedlings and resuspended in WI solution (4 mM MES (pH 5.7), 0.5 M mannitol and 20 mM KCl)[37] supplemented with 10 g l$^{-1}$ sucrose. Protoplast suspension was transferred to wells in a 6-well plate and mixed with free iP or *tZ* (2 μM), or with Sepharose beads with attached *tZ* or iP (10 μmol l$^{-1}$). Corresponding controls, without added cytokinin, were also examined with and without beads. The Sepharose bead stocks with or without attached *tZ* or iP were stored in 20% ethanol, and were washed with WI solution prior to transfer into the wells. For the treatment of *TCSn::GFP* and *TCS::GFP* protoplasts, the beads were washed twice with 1 ml WI solution. For the treatments of *TCSn::GFP ahk2,3, TCSn::GFP ahk2,4* and *TCSn::GFP ahk3,4* protoplasts and for tests of ligand detachment from beads in absence of cells, four additional washing steps were included. Samples for RNA isolation were frozen in liquid nitrogen. Samples for confocal microscopy were incubated in darkness at room temperature under continuous shaking (46 rpm) for 16 h. Five minute prior to confocal imaging, 1 μM of the dye FM4-64[42] was added to stain all cells. Twenty microlitre of the sample was used for imaging and the rest was frozen in liquid nitrogen and stored in −80 °C until cytokinin analysis for testing possible leakage of cytokinins from the beads.

**RNA isolation, cDNA synthesis and qRT-PCR.** For qRT-PCR, total RNA was extracted using RNeasy Plant Mini Kit (Qiagen) following the manufacturer instructions. Contaminating DNA was removed using TURBO DNA-free Kit (Invitrogen). First-strand cDNA was synthesised using iScript cDNA Synthesis Kit (Bio-Rad). The housekeeping gene *ACTIN2 (ACT2)* was used as an internal control for relative expression analysis. Four biological replicates were analysed in triplicate. Reaction mixtures (10 μl) comprised 5 μl LightCycler 480 SYBR Green I Master (Roche), 4 μl of the corresponding primer pair (1.5 μM each) and 1 μl of cDNA template. Relative quantification of gene expression data was performed using the comparative C$_T$ method (2$^{-\Delta\Delta Ct}$) on a CFX384 Touch Real-Time PCR Detection System (Bio-Rad). The primers used are the following: 5′-CTTCTC CATGGGATGTGGAT-3′ (forward) and 5′-CACGTCATCATCACCACACA-3′ (reverse) for *CRF6*, 5′-GGTGGCATACCGGGTTTAATACC-3′ (forward) and 5′-AATGTTCATAGTTTCACCACCCAAG-3′ (reverse) for *CRF3*, 5′-GCACCC TGTTCTTCTTACCG-3′ (forward) and 5′-AACCCTCGTAGATTGGCACA-3′ (reverse) for *ACT2*.

**AHK-GFP constructs for imaging.** The genomic sequences of the *AHK3* (At1g27320) and *CRE1/AHK4* (At2g01830) genes were amplified from genomic DNA of *Arabidopsis thaliana* ecotype Columbia (Col-0) using the following primers: 5′-AATGTCGACGGATGAGTCTGTTCCATGTGC-3′ (forward) and 5′-ATTGCGGCCGCGATTCTGTATCTGAAGGCGAATTG-3′ (reverse) for *AHK3* and 5′-ATTGTCGACTGATGAGAAGAGATTTTGTGTATAATAATAATGC-3′ (forward) and 5′-ATTGCGGCCGCGACGAAGGTGAGATAGGGATTAGG-3′ (reverse) for *CRE1/AHK4*, creating SalI linker sequences at the 5′ end, and NotI linker sequences at the 3′ end. The amplified PCR fragments were inserted into SalI and NotI sites of pENTR-2B-Dual (Invitrogen), yielding pENTR-2B-Dual-AHK3 and pENTR-2B-Dual-AHK4, respectively. Subsequently, enhanced GFP coding sequence was prepared using primers carrying NotI restriction site:

5′-AATGCGGCCGCACGGAGGTGGAGGTTCTATGGTGAGCAAGGGCGAG GAG-3′ (forward) and 5′-AATGCGGCCGCTTACTTGTACAGCTCGTCC ATGCCG-3′ (reverse), and the resulting PCR fragment was inserted into the unique NotI site of pENTR-2B-Dual-AHK3 or pENTR-2B-Dual-AHK4 to obtain C-terminal GFP fusions with AHK3 and CRE1/AHK4, respectively. These entry clones were recombined using the Gateway LR reaction (Invitrogen) into the p2GW7,0 vector, containing the 35S promoter[43], and the final constructs were used for protoplast transformations.

**3D AiryScan sample preparation, imaging and analysis**. Protoplasts were isolated from 4-day-old Arabidopsis root suspension culture in enzyme solution (1% cellulase (Yakult), 0.2% Macerozyme (Yakult) in B5-0.34 M Glc-mannitol solution; 4.4 g MS with vitamins, 30.5 g Glc, 30.5 g mannitol per l, adjusted to pH 5.5 with KOH) with slight shaking for 4 h, centrifuged at 1200 $g$ for 5 min. The pellet was washed with B5-0.34 M Glc-mannitol solution followed by one time wash with B5-0.28 M sucrose buffer (4.4 g l$^{-1}$ MS with vitamins, 9.6 g l$^{-1}$ sucrose, adjusted to pH 5.5 with KOH) and resuspended in B5-0.34 M Glc-mannitol solution to a final concentration of $2 \times 10^5$ protoplasts per 50 μl. Protoplasts were co-transfected with 4 μg of 35S::AHK3–GFP or 35S::AHK4–GFP and with 4 μg of ER marker 35S::RFP-p24δ5[44]. DNAs were gently mixed together with 50 μl of protoplast suspension and 150 μl of PEG solution [0.1 M Ca(NO$_3$)$_2$, 0.45 M mannitol, 25% PEG 6000] and incubated in the dark for 30 min. PEG was washed by adding 0.275 M Ca(NO$_3$)$_2$ solution in two steps of 500 μl each, centrifuged at 800 $g$ for 7 min, then removing 240 μl of supernatant. The protoplast pellet was resuspended in 300 μl of B5-0.34 M Glc-mannitol solution and incubated for 12 h in the dark at room temperature. Protoplasts were transferred to 35 mm glass bottom MatTek dishes (coverslip thickness #1.5) coated with poly-L-lysine (Sigma) and imaged using a Zeiss LSM 880 inverted fast Airyscan microscope with a Plan-Apochromat ×63 NA 1.4 oil immersion objective. Ten to 13.25 μm thick z-stacks of transformed cells were taken using Nyquist sampling steps. Images were then subjected to Airyscan processing. The channels were checked for correct alignment. The ER marker channel was then filtered with a Gaussian blur and converted to a mask in Fiji[45]. A custom Matlab script then determined the percentage signal present within and outside of the masked region in the channel of interest.

**Confocal microscopy**. GFP expression patterns in 6-day-old seedlings or isolated protoplasts of the transgenic Arabidopsis lines Col-0, ahk3, ahk4, ahk2, ahk2,3, ahk2,4 and ahk3,4 carrying TCSn::GFP, TCS::GFP, AHK3::GFP or AHK4::GFP were recorded using confocal laser scanning microscopy (Zeiss LSM780). The 488 nm laser line was employed for the GFP and FM4-64[42] fluorescence detection, and emission was detected between 490 and 580 nm and between 620 and 670 nm, respectively. Two tile scans were performed for root imaging and 5 × 5 tile scans for protoplast imaging.

**Semi-automated quantification using computer vision algorithm**. ImageJ (http://imagej.nih.gov/ij/) was used to quantify GFP fluorescence intensity. Fluorescence profiles of the stele and the full root were extracted using the Plot profile function. Plot profiles represent quantification from ten roots per treatment and two independent experiments were performed. For the calculation of overlap coefficients (Figs. 3a and 4a and Supplementary Fig. 6b) in treated protoplasts, semi-automated digital processing was performed with a semi-automated digital processing pipeline using iteration of morpho-mathematic filters within ImageJ. Raw images have been converted into 8-bit. Noise was reduced using a Median filter. FM4-64 channel was converted to binary and Fill Holes function was used to obtain the surface of the protoplasts as almost perfect circular structures. Protoplasts were counted and extracted from cellular debris using surface and circularity thresholds using the Analyze Particle function. Quantification of the GFP fluorescence in protoplasts was performed using the ImageJ plugin JACoP. Co-localisation coefficient corresponds to the Pearson's correlation coefficient between the two channels (FM4-64 and GFP). The GFP signal overlapping the protoplast reference surface was measured using the Manders' overlap coefficient (M2) and as an additional control on the reference values, the coverage of the protoplast on the GFP signal was also measured (M1) (Supplementary Fig. 6).

**Generation of cytokinins attached to Sepharose beads**. iP and tZ ligands possessing short linkers at the N9 position were synthesised according to the scheme in Supplementary Fig. 5, and confirmed by $^1$H and $^{13}$C NMR spectra as well as by MS and MS/MS data (Supplementary Fig. 11). Ligands were coupled to NHS-activated Sepharose$^{TM}$ 4 Fast Flow beads (GE Healthcare, United Kingdom). Control beads blocked with ethanolamine were prepared in the same way, omitting the ligand immobilisation step. Absorbance at 272 nm was used to determine the concentration of the immobilised cytokinin ligand. Full details are in Supplementary methods.

**Reporting summary**. Further information on research design is available in the Nature Research Reporting Summary linked to this article.

## Data availability
All data generated or analysed during this study are included in this published article and its Supplementary information files. Source data are provided with this paper.

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

## Acknowledgements
We thank Bruno Müller and Aaron Rashotte for critical discussions and provision of plant lines used in this work, Roger Granbom and Tamara Hernández Verdeja (UPSC, Umeå, Sweden) for technical assistance and providing materials, Zuzana Pěkná and Karolina Wojewodová (CRH, Palacký University, Olomouc, Czech Republic) for help with cytokinin receptor binding assays, and David Zalabák (CRH, Palacký University, Olomouc, Czech Republic) for provision of vector pINIIIΔEH expressing CRE1/AHK4.

The bioimaging facility of IST Austria, the Swedish Metabolomics Centre and the IST Austria Bio-Imaging facility are acknowledged for support. The work was funded by the European Molecular Biology Organization (EMBO ASTF 297-2013) (I.A.), Development —The Company of Biologists (DEVTF2012) (I.A.; C.T.), Plant Fellows (the International Post doc Fellowship Programme in Plant Sciences, 267423) (I.A.; K.L.), the Swedish Research Council (621-2014-4514) (K.L.), UPSC Berzelii Center for Forest Biotechnology (Vinnova 2012-01560), Kempestiftelserna (JCK-2711) (K.L.) and (JCK-1811) (E.-M.B., K.L.). The Ministry of Education, Youth and Sports of the Czech Republic via the European Regional Development Fund-Project "Plants as a tool for sustainable global development" (CZ.02.1.01/0.0/0.0/16_019/0000827) (O.N., O.P., R.S., V.M., L.P., K.D.) and project CEITEC 2020 (LQ1601) (M.P., J.H.) provided support, as did the Czech Science Foundation via projects GP14-30004P (M.P.) and 16-04184S (O.P., K.D., O.N.), Vetenskapsrådet and Vinnova (Verket för Innovationssystem) (T.V., S.R.), Knut och Alice Wallenbergs Stiftelse via "Shapesystem" grant number 2012.0050. A.J. was supported by the Austria Science Fund (FWF): I03630 to J.F. The research leading to these results received funding from European Union's Horizon 2020 programme (ERC grant no. 742985) and FWO-FWF joint project G0E5718N to J.F.

## Author contributions
I.A., C.T. and K.L. conceived the project; I.A. performed most of the experiments; I.A., O.N., L.P. and M.K. conducted the purification and quantification of cytokinins; I.A. and S.R. discussed and performed the confocal experiments; T.V. and I.A. developed and trained semi-automated algorithms for confocal image quantification; R.S., V.M. and K.D. developed and produced the cytokinins attached to Sepharose beads; M.P. and J.H. provided the homozygous lines of *ahk* mutant combinations with *TCSn::GFP* and *TCS:: GFP*; E-M.B. did the qRT-PCR assays; I.A. and K.O. did the apoplastic fluid experiments; O.P. generated the *AHK::GFP* constructs and Z.G., A.J. and J.F. performed the respective transfection assays and protoplast imaging. I.A., C.T. and K.L. analysed and interpreted the data; I.A. and O.N. made the figures; I.A., C.T. and K.L. wrote the paper.

## Competing interests
The authors declare no competing interests.
