## [Peer Review File · Nature Communications]

Reviewers' comments:

Reviewer #1 (Remarks to the Author):

This report re-verifies that AHKs would appear to be at the ER membrane suggesting that the receptor at the PM is able to function, at least in terms of TCSn-GFP output, presumably through binding to AHK receptors on the PM and then traditional TCS signaling to activate TCSn-GFP. There is really no physiological or transcript responses, but a lot of GFP reporter output as a proxy for response.

While I don't question the findings suggested here that there is TCS pathway signaling that starts at the PM in at least some cells of the root tip, the question is how important is that system in standard CK signaling and response. Part of why I, and I would assume others believed that at least some CK signaling must occur that way is that it would seem very odd to have the receptors at least partially localized to the PM yet be non-functional. No other changes would then be needed in TCS protein localization for this to work as AHPs are still in the cytosol and RRs in the nucleus. Clearly it is important to show that this signaling does occur, which it appears to do in some cells, but I am not sure more than that has been shown, and whether this is an essential part of plant survival is not addressed.

The overall point of this manuscript is to show that cytokinin (CK) can be perceived by AHK receptors at the plasma membrane (PM). This is accomplished using the cytokinin-responsive GFP reporter line TCSn-GFP, FACS of protoplasts, CK measurements, and with a bead attached CK that are unable to enter cells. There has been debate about how CK signaling or the TCS pathway functions, since the discovery of the CK AHK receptor localization primarily to the ER membrane, rather than the PM where it was previously thought to solely exist.

Points that should be addressed in this manuscript going forward.

1. The authors find overall increased CK levels +2x in TCSn + vs - sorted cells. How is this different than any other simple localization pattern in the root tip with a similar pattern, for example a root tip specific marker? The authors have previously (Antoniadi et al, 2015 Plant Cell) shown that distinct regions of the root tip have different levels of CK, such that this could just be correlative evidence with TCSn-GFP as being in the same region that has higher CK levels. Wouldn't exogenous treatment of the root, expanding the TCS expression pattern, followed by new GFP sorting and CK measurements be more conclusive? While the INCYDE treatment partially addresses this there should at least be some discussion of the previous authors work in relation to this.
2. The examination of FACS +/-GFP populations appears relatively clear to set sorting windows (Fig 1). However, the GFP Min and Max windows appear largely arbitrary for the dividing line between these and do not appear to cover the same, full +GFP overall window (especially at the lower end). As such it is difficult to know what to make of this data. Is a cell at the lower end of the max window really that different from the upper end of the min window? Really this seems to be an attempt to make a qualitative sensor into a quantitative one. Much more work needs to be done to show that than what is offered here. Since there are no stats on these results in Fig 1d, they should be removed and the rest of Fig1 merged into Fig 2
3. The described age of all plant roots methods in this paper is about 50/50 8d vs 9d, please correct.
4. The title of Fig 2 is misleading, based on all we know about CK and from the results shown in this figure and it should be changed. One could make a strong case just from the data in this figure that conjugates (and even non-tZ forms) are what are regulating CK response.
5. It is unclear to me how INCYDE is really functioning in this system. The chemical should block CKXs, but are all CKXs blocked to the same degree? CKXs can be organelle specific in their localization, how does that affect what is happening and what is known of INCYDE activity in Arabidopsis roots? It would seem that an easier way to see changes in CK or TCSn levels would have been to just treat roots with exogenous CK and determine the effects.

6. Some of the points raised by the authors from the INCYDE results, suggest that iP is not really involved in the responses seen, potentially not effecting TCS-GFP, yet iP is highly used in later figures in involving TCS-GFP as a CK. This should be resolved. Also, is it not hard to image that iP CK levels are not changed after the addition of INCYDE for other reasons than suggested, for example that INCYDE is not specific for iP targeted CKXs, or that iP levels after INCYDE application are backed up in the pathway, rapidly increased, and then shunted to iP9G that was found to be significantly increased in the measurements.
7. It would appear that some Figures in this paper come from work done during the First authors thesis, since it can be found on-line with highly similar figures. I have no issue with that, and think it is great to see thesis work published at any stage. ***However, there are two issues that are a problem in comparing Fig 2a (this work) and Fig 30 (thesis). One is that high cZ9G levels have been removed from the manuscript version. The second, and more important is that tZOG levels in this manuscript work are high and similar between mock and INCYDE, while in the thesis there is a minimum 10X increase in tZOG levels + INCYDE. These findings must be addressed regarding the accuracy of this work.
8. There is a lot of reliance on TCS as a cytokinin responsive element, but how many different CK forms is this reporter line really responding to? How well has it be shown that TCS represents different CK form output, is unclear to me but should at least be stated if known.
9. The axis describing the data presented in Fig 3a and 4a are unclear to me.
10. I am unclear where the iP and tZ values are significantly different in the transition zone region in Fig 3f.
11. There are extra yellow arrows in fig 4a
12. Expression shown in Fig 4C would seem to highly depend on what specific root tissues these protoplasts came from in regards to whether there should be higher or lower CK levels and PM or ER localized receptors. Please address
13. How does a bead attached CK fit into the CK-receptor binding pocket and function? Can bead-CK be competitively inhibited?
14. It would be great to see some use of the bead attached CK in a CK functional assay. Basically working as a cytokinin. This gets at a bigger question, which is how important is CK signaling from the PM vs the ER. It would be nice to hear some speculation on this point.
15. What happens to TCSn-GFP plants when they are treated with bead attached CK?

Reviewer #2 (Remarks to the Author):

Cytokinins are essential phytohormones. While the mechanistics of signal transduction has been elucidated, the subcellular compartments relevant for signal sensing are debated. Here, the authors directly measure the extra- and intracellular distribution of bioactive cytokinins and metabolites in Arabidopsis roots. For intracellular cytokinins, they identify a positive correlation between signal perception and cytokinin contents using FACS to separate cytokinin perceiving cells from non-responsive cells. When the authors determined extracellular cytokinin content, they found similar or even larger amounts of bioactive cytokinins, in the apoplast. This prompted the authors to test whether apoplastic cytokinins could initiate signaling. To test for this, bioactive iP and tZ were covalently attached to sepharose beads, which prevents cellular uptake. These modified cytokinins were able to trigger the signaling response, providing compelling functional evidence for the functional role of apoplastic cytokinin in initiating a signaling response. Importantly, the fraction of free cytokinins released from the sepharex-bound population was negligible, corroborating the relevance of extracellular cytokinins. Next, the authors set out to determine the relative contribution of the different cytokinin receptors in perceiving extracellular or free cytokinins by using various receptor mutants. This revealed that all receptors can sense extracellular cytokinins. Notably, AHK2 responded to both iP and tZ, while AHK3 showed selectivity for iP and AHK4 for tZ.

Taken together, the authors convincingly demonstrate that apoplastic cytokinins initiate signaling, at least in the root. This model of signalling is independently corroborated by the data presented in

the accompanying manuscript. The manuscript is concisely written, the experiments well designed and executed, and the conclusions supported by the experiments.

I have noticed some points that I feel the authors should address:

The beginning of the manuscript describes the intracellular cytokinin content and its correlation in cytokinin-perceiving cells. Later, the topic switches to the apoplastic cytokinins, which is also the main topic of the functional analyses. There is little further discussion or functional interpretation about the intracellular cytokinin distribution and its positive correlation with cytokinin-responsive cells.

In the discussion the authors could cite an earlier article where evidence for the role of extracellular cytokinins in *Physcomitrella* was presented:

von Schwartzenberg, K. et al. (2007). Cytokinins in the bryophyte *Physcomitrella patens*: analyses of activity, distribution, and cytokinin oxidase/dehydrogenase overexpression reveal the role of extracellular cytokinins. *Plant Physiol.* 145: 786–800.n

Reviewer #3 (Remarks to the Author):

In this manuscript, the authors examined cellular sites of cytokinin perception in root. They first analyzed cytokinin distribution by utilizing FACS and LC-MS, specifically, correlation with TCS::GFP expression and symplastic and apoplastic fractions. Then, they showed occurrence of cytokinin response even when immobilized cytokinins (bound iP, tZ) were applied. They carefully checked purity and stability of the immobilized cytokinins to exclude artificial effects. They further analyzed microscopically the localization of AHK cytokinin receptor, and detected on the cell surface. In addition to the functionality of plasma-membrane localized receptor, they provided some data suggesting differential action of iP and tZ on root based on the difference in TCS::GFP signal. While it has been clearly shown that cytokinins can be perceived in the extracellular space, differences in the effects of tZ and iP are not clear. Several major points should be addressed.

1: The ligand affinity of cytokinin receptors has been examined by various methods, and it has been shown that the affinity differs depending on the receptor. How do the authors' results relate to the findings of these previous studies?

2: The resolution of Figs 3a and 4b is not so high. Especially, Fig 4b is difficult to discuss the difference quantitatively. Supplementary Fig 6 too.

3: When iP is exogenously applied, some part of the compound is expected to be converted to tZ. What part of applied iP is converted to tZ in the author's experimental condition? Also, if the authors aim to know the difference between the action of tZ and iP more clearly, they should conduct the experiment of Fig. 3 and 4 with the mutant of *cyp735a1cyp735a2* in which conversion of iP to tZ is impaired.

4: In cytokinin quantification data, the nucleotide forms were not analyzed although they defined "total cytokinin". In general, the nucleotide pool occupies substantial part of total cytokinin-related compounds, and your SI-labeled tracer experiment suggests large pool of cytokinin metabolism (Supplementary Fig 3). Inclusion of nucleotide form data would give you useful information for interpretation of cytokinin distribution.

5: Line 102: "severe root defects in *abcg14* mutants where tZ transport is impaired."

So far as I checked the two papers, "severe root defect" was not described, and the root growth phenotype was not coincided in the two. In addition, transport substrate of ABCG14 has not been fully identified. So, this sentence should be modified appropriately.

6: Although the authors used super-resolution 3D Airyscan images, Fig 4c is still difficult to see AHK-only signal on the cell surface. More magnified images should be provided. Fig 4d label is missing.

Reviewer #4 (Remarks to the Author):

Antoniadin et al., have shown cytokinin perception and response in plant root cells is at the PM. They employed TCSn-GFP as a cytokinin reporter to indicate both the level of active cellular cytokinin and its subcellular distributions. In addition, they further compared the bioactivity of iP and tZ in cell sensing and responding to the extracellular CKs, and found that tZ plays a dominant role in this process. In the end, the CK responses and subcellular localizations of different AHKs are studied in plant roots and found that AHK4 also localized to non-ER regions which may indicate CK receptor has multiple localizations in plants. This study suggests that CK perception and signaling in plants is not only limited in ER. However, some of the conclusions based on the results are over interpreted or misinterpreted in the manuscript. Some figures need to be further improved, explained and discussed. Please see the detailed concerns below:

1. The novelty and biological significance of the study are suggested to be included at the end of the abstract rather than introducing augments and stating the complex fact of regulatory network of biological signaling pathways, which usually is (line 45-50).

2. The iP and tZ contents in root protoplast cells in Fig.1d and Fig.2a is not consistent. There are no significant differences in Fig.1d when comparing the iP and tZ contents. The conclusion made in line 95-98 is poorly supported by Fig.2a since there is enrichment of iP in the root protoplasts according to Fig.1d and Fig. 2a, not as described in line 98.

3. The perception and activation of CK response on the PM by free and conjugated tZ and iP treatments as shown in Fig. 3 a-d indicates that the CK receptors on the PM has preference of binding different types of CKs. tZ plays a leading role in activate the CK responses rather than iP. Additionally, tZ can specific binding by AHK4 in outer membranes. These results are derived from this study. However, Kubiasová et al. used iP-NBD as CK reporter and found that it can enter the secretory pathway together with CRE1/AHK4 to reach the PM. Therefore, how to explain and make sense the differences in functional roles and responses of iP and tZ in the two independent studies?

4. It is misleading and improper to state that the CK receptor was not colocalized with ER marker in line 194-197. AHK3 and AHK 4 in Fig.4 did colocalize with ER marker at majority level. However, except for the colocalized signals, they also showed a few non-colocalization signals which could represent for the additional subcellular localization of cytokinin receptors. In addition, calculation of the colocalization ratio is suggested to be included to better illustrate the result.

5. In some figures, the obtained results are not well explained and discussed in the text. It makes the manuscript difficult to understand when looking at the results. For example, the green bars in Figure 2a and b.

6. It is difficult to tell the differences in Figure 3e, where exactly in the root? Stronger or newly induced? Enlargement the arrow pointed areas in Figure 3e could be more informative and clear to the authors.

7. In figure 4a, the yellow arrows at the bottom side should be removed.

8. The labeling in Fig. 3b is difficult to follow. The authors should indicate what the red and green fluorescent signals represent for in the figure.

Editor/reviewer comments	Author responses
Editor main request We feel that new experimental data to demonstrate the physiological relevance of PM-derived CK signaling would greatly strengthen the case for further consideration.	We agree, and have now added data (Fig 3g, Supp Fig. 6e; lines 182-187) showing that members of the cytokinin-specific CRF transcription factor family are upregulated in protoplasts by both free and bead-attached cytokinins. This indicates functional relevance beyond simply showing that TCSn is activated. Protoplast approaches of course limit the possibilities for developmental experiments, but in future routes to testing PM receptor functions could be for example by preventing targeting to PM
Reviewer #1 (Remarks to the Author):	
This report re-verifies that AHKs would appear to be at the ER membrane suggesting that the receptor at the PM is able to function, at least in terms of TCSn-GFP output, presumably through binding to AHK receptors on the PM and then traditional TCS signaling to activate TCSn-GFP. There is really no physiological or transcript responses, but a lot of GFP reporter output as a proxy for response.	We now show transcript evidence for CRFs as mentioned above.
While I don't question the findings suggested here that there is TCS pathway signaling that starts at the PM in at least some cells of the root tip, the question is how important is that system in standard CK signaling and response. Part of why I, and I would assume others believed that at least some CK signaling must occur that way is that it would seem very odd to have the receptors at least partially localized to the PM yet be non-functional. No other changes would then be needed in TCS protein localization for this to work as AHPs are still in the cytosol and RRs in the nucleus. Clearly it is important to show that this signaling does occur, which it appears to do in some cells, but I am not sure more than that has been shown, and whether this is an essential part of plant survival is not addressed.	We interpret this to be essentially the same as the previous point, addressed above, that the signalling pathway from receptors to target TFs is functioning. Regarding plant survival, short term protoplast experiments are not suitable for assessing this
The overall point of this manuscript is to show that cytokinin (CK) can be perceived by AHK receptors at the plasma membrane (PM). This is accomplished using the cytokinin-responsive GFP reporter line TCSn-GFP, FACS of protoplasts, CK measurements, and with a bead attached CK that are unable to enter cells. There has been debate about how CK signaling or the TCS pathway functions, since the discovery of the CK AHK receptor localization primarily to the ER membrane, rather than the PM where it was previously thought to solely exist.	
Points that should be addressed in this manuscript going forward.	
1. The authors find overall increased CK levels +2x in TCSn + vs – sorted cells. How is this different than any other simple localization pattern in the root tip with a similar pattern, for example a root tip specific marker? The authors have previously (Antoniadi et al, 2015 Plant Cell) shown that distinct regions of the root tip have different levels of CK, such that this could just be correlative evidence with TCSn-GFP as being in the same region that has higher CK levels. Wouldn't exogenous treatment of the root, expanding the TCS expression pattern, followed by new GFP sorting and CK measurements be more conclusive? While the INCYDE treatment partially addresses this there should at least be some discussion of the previous authors work in relation to this.	Antoniadi et al 2015 showed heterogeneity in CK content among different cell types. But nobody previously checked whether CK content matches CK signalling strength. Our TCSn:GFP data provide validation that TCSn readout does indeed track the CK metabolite levels - we found 3x more CK in the GFP+ cells, and that the GFP+max cells had more CK than the GFP+min cells. We did preliminary tests where TCSn:GFP was treated also with BAP. In comparison with BAP, INCYDE treatment caused stronger induction of TCSn:GFP signal (new Supp Fig. 2 added) and therefore it was chosen for the sorting experiment. Moreover, rather than adding exogenous CK, we preferred to modulate endogenous pools via INCYDE, and this led to increased tZ in GFP+ cells. Together, these data add weight to the view that TCSn can serve as a quantitative sensor. We have now added sentences (line 96-98; line 292-4) to highlight more clearly
2. The examination of FACS +/-GFP populations appears relatively clear to set sorting windows (Fig 1). However, the GFP Min and	The GFP+ max and min cells were analysed separately but not with the intent to generate a full calibration

Max windows appear largely arbitrary for the dividing line between these and do not appear to cover the same, full +GFP overall window (especially at the lower end). As such it is difficult to know what to make of this data. Is a cell at the lower end of the max window really that different from the upper end of the min window? Really this seems to be an attempt to make a qualitative sensor into a quantitative one. Much more work needs to be done to show that than what is offered here. Since there are no stats on these results in Fig 1d, they should be removed and the rest of Fig1 merged into Fig 2	curve, instead to test the hypothesis that GFP intensity shows a broad positive correlation with CK level. See also previous point. Although Fig 1d does not indicate statistically significant changes in bioactive cytokinins, possibly due to lower replication in these technically demanding experiments, it does indicate broad trends in the level of cytokinin compounds groups: Most of the active cytokinin (yellow) and their immediate precursors (blue) are somewhat enriched in the GFP+ max cells compared to cytokinin glucosyl-conjugates (in green).
3. The described age of all plant roots methods in this paper is about 50/50 8d vs 9d, please correct.	Plants were 9 days old at all cases apart from confocal experiments (6 days). Text has been corrected
4. The title of Fig 2 is misleading, based on all we know about CK and from the results shown in this figure and it should be changed. One could make a strong case just from the data in this figure that conjugates (and even non-tZ forms) are what are regulating CK response.	The majority of biochemical and functional evidence indicates that cytokinin free bases are the bioactive forms that interact most effectively with AHK receptors. We added further reference to Lomin et al 2015 (line 55-56, 61-63). However, we agree that Fig 2 title does not fully reflect the content, and have amended
5. It is unclear to me how INCYDE is really functioning in this system. The chemical should block CKXs, but are all CKXs blocked to the same degree? CKXs can be organelle specific in their localization, how does that affect what is happening and what is known of INCYDE activity in Arabidopsis roots? It would seem that an easier way to see changes in CK or TCSn levels would have been to just treat roots with exogenous CK and determine the effects.	Catalytic sites of CKXs appear to be highly conserved not only within the family members but also across different plant species (Gu et al., J Plant Growth Regul (2010) 29:428–440). so we predict that INCYDE will block all to approximately similar degrees. The fact that INCYDE influences bioactive CK pool sizes and TCSn readout in the same direction (both enhanced) provides evidence that this perturbation of the system results in dynamic changes to CK signalling
6. Some of the points raised by the authors from the INCYDE results, suggest that iP is not really involved in the responses seen, potentially not effecting TCS-GFP, yet iP is highly used in later figures in involving TCS-GFP as a CK. This should be resolved. Also, is it not hard to image that iP CK levels are not changed after the addition of INCYDE for other reasons than suggested, for example that INCYDE is not specific for iP targeted CKXs, or that iP levels after INCYDE application are backed up in the pathway, rapidly increased, and then shunted to iP9G that was found to be significantly increased in the measurements.	This is a slight misunderstanding of the data. We show that exogenous IP can induce a response from both outside and inside cells, but the evidence on endogenous levels shows no big changes and no correlation with GFP strength. We agree that IP nonetheless is an important bioactive cytokinin.
7. It would appear that some Figures in this paper come from work done during the First authors thesis, since it can be found on-line with highly similar figures. I have no issue with that, and think it is great to see thesis work published at any stage. ***However, there are two issues that are a problem in comparing Fig 2a (this work) and Fig 30 (thesis). One is that high cZ9G levels have been removed from the manuscript version. The second, and more important is that tZOG levels in this manuscript work are high and similar between mock and INCYDE, while in the thesis there is a minimum 10X increase in tZOG levels + INCYDE. These findings must be addressed regarding the accuracy of this work.	There was an error in calibration curve calculations for cZ9G and tZOG in the thesis. The corrected values are presented in the manuscript.
8. There is a lot of reliance on TCS as a cytokinin responsive element, but how many different CK forms is this reporter line really responding to? How well has it be shown that TCS represents different CK form output, is unclear to me but should at least be stated if known.	Muller & Sheen (2008, Nature) showed that a range of native and synthetic cytokinins all activate TCS. Our work with tZ and IP is consistent with their findings, and we have added a phrase to this effect (line 190)
9. The axis describing the data presented in Fig 3a and 4a are unclear to me.	We have re-worded for added clarity.
10. I am unclear where the iP and tZ values are significantly different in the transition zone region in Fig 3f.	There are significant zone-specific effects, but error bars visually obscured by the close packed data points. We have replotted to better show the error bars We also added further details in legend and Methods.

11. There are extra yellow arrows in fig 4a	Spurious arrows removed from Fig 4
12. Expression shown in Fig 4C would seem to highly depend on what specific root tissues these protoplasts came from in regards to whether there should be higher or lower CK levels and PM or ER localized receptors. Please address	We agree that different cell types will likely have different PM vs ER receptor proportions, and in Antoniadis et al 2015 we already showed the heterogeneity of CK content. Here we only aimed to test whether some AHKs are on PM, rather than providing comprehensive evaluation of the relative proportions between PM and ER. Discussion added on this point (line 300-3) and quantitative analysis added as Fig 4d.
13. How does a bead attached CK fit into the CK-receptor binding pocket and function? Can bead-CK be competitively inhibited?	The N9-linked spacer allows flexibility and is predicted to cause minimal steric hindrance. The generally lower response to bead CKs compared with equivalent free CK concentration suggests nonetheless that bead CKs are somewhat less active, while still being capable of eliciting highly significant TCS responses compared with non-CK control beads
14. It would be great to see some use of the bead attached CK in a CK functional assay. Basically working as a cytokinin. This gets at a bigger question, which is how important is CK signaling from the PM vs the ER. It would be nice to hear some speculation on this point.	Addressed above (new data and text added), see first point made by Editor. We also contend that the TCSn readout is a functional rapid and sensitive assay conducted in living cells, as opposed to in vitro studies.
15. What happens to TCSn-GFP plants when they are treated with bead attached CK?	We did not do such experiments, but predict that nothing will happen because cell walls will prevent access to PM receptors.
Reviewer #2 (Remarks to the Author):	
Cytokinins are essential phytohormones. While the mechanistic of signal transduction has been elucidated, the subcellular compartments relevant for signal sensing are debated. Here, the authors directly measure the extra- and intracellular distribution of bioactive cytokinins and metabolites in Arabidopsis roots. For intracellular cytokinins, they identify a positive correlation between signal perception and cytokinin contents using FACS to separate cytokinin perceiving cells from non-responsive cells. When the authors determined extracellular cytokinin content, they found similar or even larger amounts of bioactive cytokinins, in the apoplast. This prompted the authors to test whether apoplastic cytokinins could initiate signaling. To test for this, bioactive iP and tZ were covalently attached to sepharose beads, which prevents cellular uptake. These modified cytokinins were able to trigger the signaling response, providing compelling functional evidence for the functional role of apoplastic cytokinin in initiating a signaling response. Importantly, the fraction of free cytokinins released from the sepharex-bound population was negligible, corroborating the relevance of extracellular cytokinins. Next, the authors set out to determine the relative contribution of the different cytokinin receptors in perceiving extracellular or free cytokinins by using various receptor mutants. This revealed that all receptors can sense extracellular cytokinins. Notably, AHK2 responded to both iP and tZ, while AHK3 showed selectivity for iP and AHK4 for tZ.	We appreciate these positive comments
Taken together, the authors convincingly demonstrate that apoplastic cytokinins initiate signaling, at least in the root. This model of signalling is independently corroborated by the data presented in the accompanying manuscript. The manuscript is concisely written, the experiments well designed and executed, and the conclusions supported by the experiments.	
I have noticed some points that I feel the authors should address:	
The beginning of the manuscript describes the intracellular cytokinin content and its correlation in cytokinin-perceiving cells. Later, the topic switches to the apoplastic cytokinins, which is also the main topic of the functional analyses. There is little further discussion or functional interpretation about the intracellular	We agree with this point, and have added further depth to discussion of intracellular cytokinins (lines 280-294)

cytokinin distribution and its positive correlation with cytokinin-responsive cells.	
In the discussion the authors could cite an earlier article where evidence for the role of extacellular cytokinins in Physcomitrella was presented: von Schwartzberg, K. et al. (2007). Cytokinins in the bryophyte Physcomitrella patens : analyses of activity, distribution, and cytokinin oxidase/dehydrogenase overexpression reveal the role of extracellular cytokinins. Plant Physiol. 145: 786–800	We agree that this is a valuable relevant source, now added and discussed (line 311-313), also we now included Motyka et al. 2003 on secreted cytokinins in suspension cell cultures.
Reviewer #3 (Remarks to the Author):	
In this manuscript, the authors examined cellular sites of cytokinin perception in root. They first analyzed cytokinin distribution by utilizing FACS and LC-MS, specifically, correlation with TCS::GFP expression and symplastic and apoplastic fractions. Then, they showed occurrence of cytokinin response even when immobilized cytokinins (bound iP, tZ) were applied. They carefully checked purity and stability of the immobilized cytokinins to exclude artificial effects. They further analyzed microscopically the localization of AHK cytokinin receptor, and detected on the cell surface. In addition to the functionality of plasma-membrane localized receptor, they provided some data suggesting differential action of iP and tZ on root based on the difference in TCS::GFP signal. While it has been clearly shown that cytokinins can be perceived in the extracellular space, differences in the effects of tZ and iP are not clear. Several major points should be addressed.	
1: The ligand affinity of cytokinin receptors has been examined by various methods, and it has been shown that the affinity differs depending on the receptor. How do the authors' results relate to the findings of these previous studies?	We didn't cover this aspect in depth because we were not reporting affinity data. But we have added some more discussion (line 334-7)
2: The resolution of Figs 3a and 4b is not so high. Especially, Fig 4b is difficult to discuss the difference quantitatively. Supplementary Fig 6 too.	We repeated the image analysis and processed the stele regions separately, now shown as insets in Fig. 3e and 4b to emphasise that the main response to added cytokinin is in stele rather than columella. Description expanded (line 190-8)
3: When iP is exogenously applied, some part of the compound is expected to be converted to tZ. What part of applied iP is converted to tZ in the author's experimental condition? Also, if the authors aim to know the difference between the action of tZ and iP more clearly, they should conduct the experiment of Fig. 3 and 4 with the mutant of cyp735a1cyp735a2 in which conversion of iP to tZ is impaired.	We now quantified tZ in samples treated with free iP in our experimental set up (16 h treatment of TCSn:GFP root protoplasts), new data added (Supp Table 1) described line 193-5. The conclusion is that no significant additional tZ was found following supply of iP, and therefore the activity attributed to exogenous iP is a correct interpretation rather than reflecting substantial conversion to tZ.
4: In cytokinin quantification data, the nucleotide forms were not analyzed although they defined "total cytokinin". In general, the nucleotide pool occupies substantial part of total cytokinin-related compounds, and your SI-labeled tracer experiment suggests large pool of cytokinin metabolism (Supplementary Fig 3). Inclusion of nucleotide form data would give you useful information for interpretation of cytokinin distribution.	We agree that metabolism to and from nucleotide forms may be important. In our experiments not all nucleotide forms were consistently detected, possibly because of fast turnover and/or matrix effects during LCMS. We clarified in Fig 1 legend to indicate the values are sum of detected cytokinins.
5: Line 102: "severe root defects in abcg14 mutants where tZ transport is impaired." So far as I checked the two papers, "severe root defect" was not described, and the root growth phenotype was not coincided in the two. In addition, transport substrate of ABCG14 has not been fully identified. So, this sentence should be modified appropriately.	Root growth defects were found in abcg14 of Arabidopsis (Zhang 2014) and abcg18 of rice (https://academic.oup.com/jxb/article/70/21/6277/5554342), although interestingly Ko 2014 showed no defect. Rice ABCG18 has ability to enhance export, shown to depend on cytokinin type. Reference added and text modified (line125-7).
6: Although the authors used super-resolution 3D Airyscan images, Fig 4c is still difficult to see AHK-only signal on the cell surface. More magnified images should be provided. Fig 4d label is missing.	We modified the arrows in Fig 4c to better highlight surface green zones and punctae, and added further images in new Supp Fig 9. We believe the images now make a convincing case for AHKs at the cell surface.

Reviewer #4 (Remarks to the Author):	
Antoniadin et al., have shown cytokinin perception and response in plant root cells is at the PM. They employed TCSn-GFP as a cytokinin reporter to indicate both the level of active cellular cytokinin and its subcellular distributions. In addition, they further compared the bioactivity of iP and tZ in cell sensing and responding to the extracellular CKs, and found that tZ plays a dominant role in this process. In the end, the CK responses and subcellular localizations of different AHKs are studies in pant roots and found that AHK4 also localized to non-ER regions which may indicates CK receptor has multiple localizations in plants. This study suggests that CK perception and signaling in plants is not only limited in ER. However, some of the conclusions based on the results are over interpreted or misinterpreted in the manuscript. Some figures need to be further improved, explained and discussed. Please see the detailed concerns below:	
1. The novelty and biological significance of the study are suggested to be included at the end of the abstract rather than introducing augments and stating the complex fact of regulatory network of biological signaling pathways, which usually is (line 45-50).	We removed the last part of the abstract to reflect this point
2. The iP and tZ contents in root protoplast cells in Fig.1d and Fig.2a is not consistent. There are no significant differences in Fig.1d when comparing the iP and tZ contents. The conclusion made the in line 95-98 is poorly supported by Fig.2a since there is enrichment of iP in the root protoplasts according to Fig.1d and Fig. 2a, not as described in line 98.	There is a small misunderstanding here. These are not the same data type: Fig 1d is the max/min ratio within the GFP+ population, and Fig 2a is the overall GFP+/-ratio. iP was not enriched in the total GFP+ cells (Fig2a) whereas tZ was enriched. Text modified to clarify (line 120-5)
3. The perception and activation of CK response on the PM by free and conjugated tZ and iP treatments as shown in Fig. 3 a-d indicates that the CK receptors on the PM has preference of binding different types of CKs. tZ plays a leading role in activate the CK responses rather than iP. Additionally, tZ can specific bingding by AHK4 in outer membranes. These results are derived from this study. However, Kubiasová et al. used iP-NBD as CK reporter and found that it can enter the secretory pathway together with CRE1/AHK4 to reach the PM. Therefore, how to explain and make sense the differences in functional roles and responses of iP and tZ in the two independent studies?	We agree that tZ and iP can have different results in different contexts, and likely also depending on methods used, and on whether exogenous or endogenous compounds are being studied. As Kubiasova et al. did not test the tZ equivalent of IP-NBD, we don't know whether that approach would reveal a similar role for tZ. We feel it is unwise to speculate too much on this point, but have added a small discussion (line 326-332)
4. It is misleading and improper to state that the CK receptor was not colocalized with ER maker in line 194-197. AHK3 and AHK 4 in Fig.4 did colocalize with ER marker at majority level. However, except for the colocalized signals, they also showed a few non-colocalization signals which could represent for the additional subcellular localization of cytokinin receptors. In addition, calculation of the colocalization ratio is suggested to be included to better illustrate the result.	This may be a misunderstanding of our text. We say " proportion of both CK receptors was not colocalised with ER", and it is certainly the case that substantial AHKs are on ER. New quantitative data added (Fig 4d) to further emphasise the point
5. In some figures, the obtained results are not well explained and discussed in the text. It makes the manuscript difficult to understand when looking at the results. For example, the green bars in Figure 2a and b.	We are not sure where the confusion lies here. Fig 2 defines all colours in parts a and c, and also mentioned in the legend. But we have added some more description of colour coding (line 120, 125).
6. It is difficult to tell the differences in Figure 3e, where exactly in the root? Stronger or newly induced? Enlargement the arrow pointed areas in Figure 3e could be more informative and clear to the authors.	New images now provided
7. In figure 4a, the yellow arrows at the bottom side should be removed.	Done
8. The labeling in Fig. 3b is difficult to follow. The authors should indicate what the red and green florescent signals represent for in the figure.	Legend expanded to indicate this better

Reviewers' comments:

Reviewer #3 (Remarks to the Author):

The authors have addressed some of my criticisms. However, other points still remain to be addressed.

The data of Figure 4 does not match the legends. No improvement.

The authors answered that they did not include the nucleotide-type precursors of cytokinin to results, and explained that they modified Figure 1 legend as "sum of detected cytokinins". However, as far as I checked, they have not been corrected.

Lines 125 to 127: The authors describe the results of the Arabidopsis *abcg14* mutant on root growth and the results of the rice *abcg18* mutant comparably, but they should be more careful. Xylem sap cytokinins were analyzed in both mutants. In the results, the overall cytokinin concentration was greatly reduced in *abcg14*, whereas the concentration of tZ-type was reduced but that of iP-type was rising in *abcg18*. The transporters involved in the loading of cytokinins into xylem in Arabidopsis and rice do not always have the same substrate specificity. I suggest the authors to delete the sentence.

I could not evaluate the improvement of Fig 4c because of lacking the data. The image of Suppl Fig. 9 also does not show localization AHK-signal so clearly. If possible, the authors should show clearer images.

Reviewer #4 (Remarks to the Author):

The authors have improved the manuscript basing on my comments and suggestions. They added new discussion or description in the text to make further clarifications and also modified their Figures accordingly. Therefore, I have no additional comments, and I would like to recommend it for publication.

Reviewer #5 (Remarks to the Author):

This is interesting work and elegantly demonstrated the perception of CK by AHK receptors at the plasma membrane. In the view of a bioorganic chemist, I still have concern for the perception of the immobilized CK analogs at PM-localized AHK.

One of key experimental results supporting the conclusion is that Sepharose-bound cytokinin ANALOGs (N9-substituted derivative of iP and tZ) could elicit the TCS activation via plasma membrane-localized AHKs.

In the experimental methods, the author added Sepharose 4 Fast flow beads-bound cytokinin analogs. The structure of a bound form of iP and tZ derivatives is as follows.

The CK analog bound beads are added to protoplast suspension at the concentration of 2 microM that is estimated by UV absorption of beads. The bound form of CK on beads can activate GFP expression in protoplast via cell surface localized AHK receptors.

I have some concern about the experimental design and control experiments.

(1) Based on the previous reports A and B indicated below, the introduction of alkyl chain/substitution would greatly reduce the cytokinin activity including AHK binding. The ligand-receptor interaction occurs by a chemical bond between two molecules (non-covalent binding). In this study, the ligand is immobilized on beads and, additionally, AHK receptor is trapped on the cell surface. The binding efficacy between these two immobilized molecules would be significantly lower than that of the immobilized ligand and the soluble receptor because the binding event can occur only when the cell surface attached with beads. In contrast, the soluble receptor (protein) can freely access to the ligand-bounded beads. Further, protoplast cells could not access the inside matrix of beads (sepharose is cross-linked agar beads). N9-linked CK also immobilized inside the matrix of beads (Figure S6).

If the N9-linked CK analogs can fully activate TCS signal in protoplast at 10-500 nanoM range, this is convincing evidence. So, I recommend that author show the CK activity of N9-linked CK analogs (reaction intermediate used in this study) on GFP reporter assay in

protoplast system.

I think this experiment is important because several papers have reported that modification of N9 position will decrease the CK activity. So, it is crucial to demonstrate that N9-linked CK derivatives of CK (intermediates) are still active at reasonable concentrations in protoplast assay system. Because this work argues that the interaction between two immobilized molecules. Especially, the ligand is a small molecule that will not exhibit a high affinity like antibody.

(A) *Phytochemistry*, 150, June 2018, Pages 1-11, Design, synthesis and perception of fluorescently labeled isoprenoid cytokinins.

B) *Bioorganic & Medicinal Chemistry* (2011), 19(23), 7244-7251. N9-Substituted N6-[(3-methylbut-2-en-1-yl)amino]purine derivatives and their biological activity in selected cytokinin bioassays)

(2) Authors measured the free tZ and iP by LC-MS in the medium after the incubation (Fig. 3, Fig. S8) to show the stability of CK-bound beads. However, these data confirmed that only the amounts of free tZ and iP released from Sepharose beads. The linker in N9-CK bound sepharose has amido bonds in its linker, and the amido bonds can be hydrolyzed to release CKs having N9-linkers. The medium after the incubation of beads / free CK with or without protoplast are filtrated /centrifuged to remove the beads and protoplasts. The recovered medium was again assayed with new protoplast. If any active CK analogs hydrolyzed from beads remain as soluble form, these can activate TCS signals in new protoplast.

(0) Nature communications require adequate data to support their assignment of identity and purity for each new compound described in the manuscript. The compounds in supplemental Figure 5 were many new compounds (not found in Scifinder registry database: Chemical Abstract Registry Data). However, there are NOT any description of synthetic methods and NMR, MS data. Please see the publishing policies on “Characterization of chemical and biomolecular materials”

Response to Reviewer Comments

Reviewer #3 (Remarks to the Author):

1. The authors have addressed some of my criticisms. However, other points still remain to be addressed. The data of Figure 4 does not match the legends. No improvement.

We apologise for this error. During final assembly I incorrectly inserted a previous version of Fig 4 that lacked parts C and D. Indeed it makes no sense. Correct Figure now added

2. The authors answered that they did not include the nucleotide-type precursors of cytokinin to results, and explained that they modified Figure 1 legend as “sum of detected cytokinins”. However, as far as I checked, they have not been corrected.

Now corrected. Legend reworded to “sum of detected cytokinin metabolites” instead of “sum of total cytokinin metabolites”, and axis title Fig 1b also changed to “sum of detected cytokinins”

3. Lines 125 to 127: The authors describe the results of the Arabidopsis abcg14 mutant on root growth and the results of the rice abcg18 mutant comparably, but they should be more careful. Xylem sap cytokinins were analyzed in both mutants. In the results, the overall cytokinin concentration was greatly reduced in abcg14, whereas the concentration of tZ-type was reduced but that of iP-type was rising in abcg18. The transporters involved in the loading of cytokinins into xylem in Arabidopsis and rice do not always have the same substrate specificity. I suggest the authors to delete the sentence.

Agree, contradictory literature does not help reader to understand our data. Sentence deleted.

4. I could not evaluate the improvement of Fig 4c because of lacking the data.

See first point above about error in Fig 4

5. The image of Suppl Fig. 9 also does not show localization AHK-signal so clearly. If possible, the authors should show clearer images.

This is now Supp Fig 10. Instead of relying on diffraction limited microscopes, we used 3D Airyscan to provide a super resolution view of the localization of the proteins. The images are at the limit of resolution, and at this magnification dynamics of proteins on membranes of live cells likely have an influence. Nonetheless the images in Fig. 4c and Supp Fig. 10 are representative of the clearest examples we obtained. For greater clarity we have re-positioned arrows to better indicate AHK specific signal at the PM. Images also recoloured to avoid colour blind combinations

Reviewer #4 (Remarks to the Author):

No queries to address

Reviewer #5 (Remarks to the Author):

This is interesting work and elegantly demonstrated the perception of CK by AHK receptors at the plasma membrane. In the view of a bioorganic chemist, I still have concern for the perception of the immobilized CK analogs at PM-localized AHK. (please see the following comments and the attached file illustrating the structures). One of key experimental results supporting the conclusion is that Sepharose-bound cytokinin ANALOGs (N9-substituted derivative of iP and tZ) could elicit the TCS activation via plasma membrane-localized AHKs. In the experimental methods, the author added Sepharose 4 Fast flow beads-bound cytokinin analogs. The structure of a bound form of iP and tZ derivatives is as follows. The CK analog bound beads are added to protoplast suspension at the concentration of 2 µM that is estimated by UV absorption of beads. The bound form of CK on beads can activate GFP expression in protoplast via cell surface localized AHK receptors. I have some concern about the experimental design and control experiments.

- (1) Based on the previous reports A and B indicated below, the introduction of alkyl chain/substitution would greatly reduce the cytokinin activity including AHK binding.

This is an important point. We agree that affinity is predicted to be lower, and indeed that is what we found with competitive binding studies using the CK+linker, data now added in Supp Fig 8d

The ligand-receptor interaction occurs by a chemical bond between two molecules (non-covalent binding). In this study, the ligand is immobilized on beads and, additionally, AHK receptor is trapped on the cell

surface. The binding efficacy between these two immobilized molecules would be significantly lower than that of the immobilized ligand and the soluble receptor because the binding event can occur only when the cell surface attached with beads. In contrast, the soluble receptor (protein) can freely access to the ligand-bounded beads.

All AHK receptors are located on membranes (ER or PM), so solubility is not relevant. Ligand immobilised on beads is of course restricted compared with equivalent ligand in free solution

Further, protoplast cells could not access the inside matrix of beads (sepharose is cross-linked agar beads). N9-linked CK also immobilized inside the matrix of beads (Figure S6).

We agree that some CK ligand will be attached internally within beads, and this will contribute to our demonstrated higher estimated concentrations needed to elicit equivalent response to free ligand. If the N9-linked CK analogs can fully activate TCS signal in protoplast at 10-500 nanoM range, this is convincing evidence. So, I recommend that author show the CK activity of N9-linked CK analogs (reaction intermediate used in this study) on GFP reporter assay in protoplast system. I think this experiment is important because several papers have reported that modification of N9 position will decrease the CK activity. So, it is crucial to demonstrate that N9-linked CK derivatives of CK (intermediates) are still active at reasonable concentrations in protoplast assay system.

We agree that evidence for activity of N9-linked CKs is important, and have added new data as Supp Fig 8a,d. The CKs with linkers at N9 position retain bioactivity in TCSn activation, and compete significantly with free CKs for receptor binding, but as predicted the activity is reduced compared with parent CKs

Because this work argues that the interaction between two immobilized molecules. Especially, the ligand is a small molecule that will not exhibit a high affinity like antibody.

We are not sure if comparison to antibody binding is necessary or fully relevant. Mainly because high CK affinity to AHK has been measured directly, published data indicating low nanomolar Kd values (Wulfetange et al. 2011 Plant Physiol; Stolz et al. 2011 Plant J)

(2) Authors measured the free tZ and iP by LC-MS in the medium after the incubation (Fig. 3, Fig. S8) to show the stability of CK-bound beads. However, these data confirmed that only the amounts of free tZ and iP released from Sepharose beads. The linker in N9-CK bound sepharose has amido bonds in its linker, and the amido bonds can be hydrolyzed to release CKs having N9-linkers. The medium after the incubation of beads / free CK with or without protoplast are filtrated /centrifuged to remove the beads and protoplasts. The recovered medium was again assayed with new protoplast. If any active CK analogs hydrolyzed from beads remain as soluble form, these can activate TCS signals in new protoplast.

We agree that cleavage to release CK+linker (rather than just free CK) is also possible. There is indeed a measurable amount (1-3%), somewhat more than that of free CK (0.2%), see new Supp. Table 1. However, once the lower activity of the CK+linker is taken into account (Supp Fig. 8a,d), neither the CK+linker nor free CK nor sum of both can account for the high TCSn signal seen. We therefore retain the original conclusion that the vast majority of the bead-induced TCSn-GFP signal is due to immobilised CK on the beads rather than detached ligands. New text inserted at lines 162-175

(3) Nature communications require adequate data to support their assignment of identity and purity for each new compound described in the manuscript. The compounds in supplemental Figure 5 were many new compounds (not found in Scifinder registry database: Chemical Abstract Registry Data). However, there are NOT any description of synthetic methods and NMR, MS data. Please see the publishing policies on "Characterization of chemical and biomolecular materials"

We added further details on characterisation of synthesised materials in Supplementary Methods, including ¹³C NMR to supplement the ¹H NMR already provided. In addition, we provide MS and MS/MS confirmation of the cytokinin-linker materials and their fragmentation patterns (new Supp Fig. 11). The source data for the construction of the chromatograms, MS and MS/MS spectra of each ligand are included in the Source Data file. We hope this now fully complies with NComms policies. The raw MS and NMR files each comprise >100Mb, so have not put in the Source Data Excel file, but can provide these separately if required.

REVIEWER COMMENTS

Reviewer #5 (Remarks to the Author):

In this version of the revised manuscript, the authors performed additional experiments and have addressed my concerns for the CK activity of N-linked CK ligands on Sepharose beads. I have some comments on this version of manuscript.

(1) In the manuscript, the concentration of bound CKs in the protoplast assay was indicated as 2 μ M that is calculated from the UV absorption of the beads suspension. Please specify the estimated ligand density of bound CK (μ mol / mL beads volume) in the main text and methods section of "Protoplast treatments". In the specification of NHS-activated Sepharose, the ligand density of NHS group was about 16-23 micro-mol NHS/ml drained medium (equivalent to 16-23 mM NHS/ gel means the maximum ca.15-20 mmol bound CK in 1 L of gel). The volume of beads and the UV absorption are given, the actual ligand density of bound CK ligands on beads (μ mol / mL beads volume) can be estimated. This value would represent the significantly high concentration of bound CK ligand around the microregion of the bead surface that is enough concentration to activate AHK. The amount of bound CK on the microregion might be higher values than those of suspended bound CK-beads (2 μ M) or free N-linker tZ ligand (10 μ M).

(2) Plant material used in Supplementary Table 1 is missing. Supplemental table1 displayed that "Cell culture treated with 1 μ M immobilised Cytokinins for 16 h" Does it mean that 1 μ M immobilised Cytokinins were treated with "protoplast" for 16h as the same condition in Fig.3b? However, there is no description on the plant materials in the method section (page 37, line 860-862). The immobilised cytokinin are cultured with the protoplast cell or the seedling or others ?. Is the experimental condition the same as in Fig. 3c? In Fig.3c, 2 μ M immobilized tZ release 3 nM free tZ after the incubation with the protoplast for 16h. Supplemental table1 indicated that 1 μ M immobilized tZ released 0.03 nM free tZ after 16h incubation. There is a large difference of the tZ and iP amounts in Fig. 3c and Table 1. Is this value correct?

RESPONSES TO REVIEWER

Reviewer #5 (Remarks to the Author):

In this version of the revised manuscript, the authors performed additional experiments and have addressed my concerns for the CK activity of N-linked CK ligands on Sepharose beads. I have some comments on this version of manuscript.

(1) In the manuscript, the concentration of bound CKs in the protoplast assay was indicated as 2uM that is calculated from the UV absorption of the beads suspension. Please specify the estimated ligand density of bound CK (umol / mL beads volume) in the main text and methods section of "Protoplast treatments". In the specification of NHS-activated Sepharose, the ligand density of NHS group was about 16-23 micro-mol NHS/ml drained medium (equivalent to 16-23 mM NHS/ gel means the maximum ca.15-20 mmol bound CK in 1 L of gel). The volume of beads and the UV absorption are given, the actual ligand density of bound CK ligands on beads (umol / mL beads volume) can be estimated. This value would represent the significantly high concentration of bound CK ligand around the microregion of the bead surface that is enough concentration to activate AHK. The amount of bound CK on the microregion might be higher values than those of suspended bound CK-beads (2uM) or free N-linker tZ ligand (10uM).

We have now added these values (lines 854-6), and have calculated them as ligand density in the actual incubations where protoplasts and beads are mixed in suspension, but also quote the undiluted density in packed beads (2 mmol/L). Note that in re-calculating ligand densities/concentrations, we found that the normal amount in bead incubations with protoplasts was 10 umol/L rather than the previously indicated 2 uM. We added explanation (line 167-170) that in addition to cytokinin-linker ligands having lower intrinsic bioactivity, some of the bead bound ligand will be internal within beads and thus not accessible to the receptors on cell surface. Therefore a moderate (5x) excess of bead CKs (10 umol/L density) over free CKs (2 uM) was used. These calculations are necessarily estimates and the real effective ligand density on bead surface is quite hard to know for certain. As mentioned in previous responses to reviewers, we remain fully confident that after taking account of all background effects (free CK or linker-CK), the AHK/TCS activation by the CKs attached to beads is highly significant.

(2) Plant material used in Supplementary Table 1 is missing. Supplemental table1 displayed that "Cell culture treated with 1 uM immobilised Cytokinins for 16 h" Does it mean that 1 uM immobilised Cytokinins were treated with "protoplast" for 16h as the same condition in Fig.3b? However, there is no description on the plant materials in the method section (page 37, line 860-862). The immobilised cytokinin are cultured with the protoplast cell or the seedling or others ?. Is the experimental condition the same as in Fig. 3c?

Unfortunately there was an editing error in the Supp Table 1 description, now corrected. This experiment was actually a deliberately simplified test to focus on ligand detachment during incubations with beads in the normal culture medium. No protoplasts or cell culture were included here because metabolism of any detached ligands by the cells would complicate interpretation.

In Fig.3c, 2uM immobilized tZ release 3 nM free tZ after the incubation with the protoplast for 16h. Supplemental table1 indicated that 1uM immobilized tZ released 0.03 nM free tZ after 16h incubation. There is a large difference of the tZ and iP amounts in Fig. 3c and Table 1. Is this value correct?

The reviewer is correct to raise this query. We have now added explanation (lines 429-423). In our experiment series reported in Fig 3c, the beads were washed 2x to remove free ligands. The measured amount of free CKs was 3-4 nM, too low to cause activation, as shown in Fig 3d and discussed already in the

m/s. However, for subsequent experiments we included more exhaustive washing (6x), and this is reflected in even lower free CK, reported in Supp Table 1.

There are two possible sources of free ligands (a) existing free ligand contained within the bead matrix or storage solutions – this will be removed by repeated washing steps. (b) newly detached ligands due to any instability in the bonds either CK-linker or linker-bead. We see constant amount of free CK (Supp Table 1) over time, indicating no new degradation is occurring, so the highly active free CKs never contribute significantly to activation. For the CK-linker, we see some additional amount appearing over time (max. 0.2-0.6%, see Supp Table 1). Because the CK-linker molecule is much less active than free CK (Supp Fig 8a,d), these amounts are also not sufficient to explain the high levels of activation by CK-beads seen in many incubations.

REVIEWERS' COMMENTS:

Reviewer #5 (Remarks to the Author):

In the revised version, the authors have appropriately addressed all my comments in this revised version of the manuscript.

Antoniadi et al. Response to reviewers

Reviewer #5 (Remarks to the Author):

In the revised version, the authors have appropriately addressed all my comments in this revised version of the manuscript.

RESPONSE: We are happy to see that no further revisions are requested